# Partial Wasserstein Adversarial Network for Non-rigid Point Set Registration

**Zi-Ming Wang, Nan Xue, Ling Lei, Gui-Song Xia**[*]
CAPTAIN
Wuhan University

## Abstract

Given two point sets, the problem of registration is to recover a transformation that matches one set to the other. This task is challenging due to the presence of large number of outliers, the unknown non-rigid deformations and the large sizes of point sets. To obtain strong robustness against outliers, we formulate the registration problem as a partial distribution matching (PDM) problem, where the goal is to partially match the distributions represented by point sets in a metric space. To handle large point sets, we propose a scalable PDM algorithm by utilizing the efficient partial Wasserstein-1 (PW) discrepancy. Specifically, we derive the Kantorovich-Rubinstein duality for the PW discrepancy, and show its gradient can be explicitly computed. Based on these results, we propose a partial Wasserstein adversarial network (PWAN), which is able to approximate the PW discrepancy by a neural network, and minimize it by gradient descent. In addition, it also incorporates an efficient coherence regularizer for non-rigid transformations to avoid unrealistic deformations. We evaluate PWAN on practical point set registration tasks, and show that the proposed PWAN is robust, scalable and performs more favorably than the state-of-the-art methods.

## 1 Introduction

Point set registration is a fundamental task in many computer vision applications, such as object tracking (Gao & Tedrake, 2018), shape retrieval (Berger et al., 2017) and contour matching (Avots et al., 2019). As illustrated in Fig. 1, given two point sets representing two partially overlapped shapes, *i.e.*, a reference set and a source set, the goal of this task is to recover an appropriate transformation that matches the source set to the reference one. It is challenging due to the following factors: 1) The point sets may contain a fraction of outliers which do not have valid correspondences in the other point set, such as the noise points and the points in the non-overlapped region of the shapes. 2) The number of points consisted in the point sets may be huge. 3) The deformation of point sets is generally unknown and can be non-rigid.

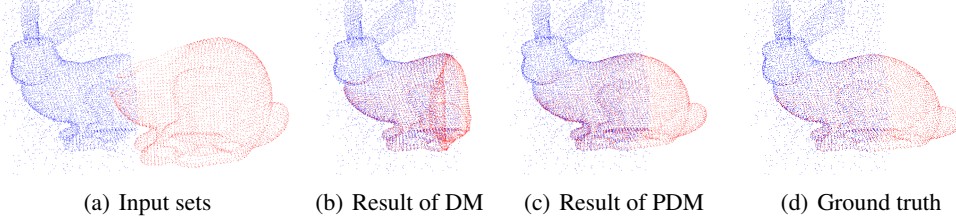

|     (a) Input sets     |     (b) Result of DM     |     (c) Result of PDM     |     (d) Ground truth     |

Figure 1: An example of non-rigid point set registration using the distribution matching (DM) formulation (Hirose, 2021a) and the proposed partial distribution matching (PDM) formulation.

The registration problem is generally solved in the distribution matching (DM) framework, where the point sets are regarded as probability distributions, and are aligned by minimizing a discrepancy between them. To reduce the influence of outliers, practical registration methods utilize the robust

---

[*]Corresponding authors

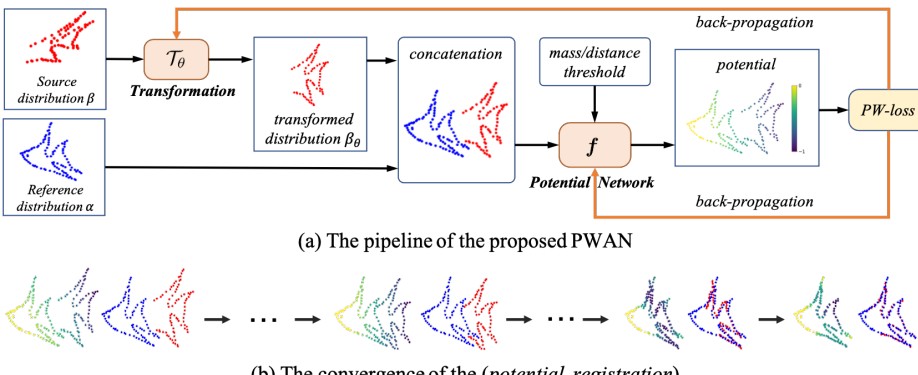

(a) The pipeline of the proposed PWAN

(b) The convergence of the (*potential, registration*)

Figure 2: An overview of PWAN. The transformation $\mathcal{T}_\theta$ and the network $f$ are trained adversarially.

discrepancies, such as Kullback-Leibler divergence (Myronenko & Song, 2010; Hirose, 2021a) and $L_2$ distance (Jian & Vemuri, 2011). Thus they are able to greedily align the largest possible fraction of points while being tolerant of a small number of outliers. However, for point sets that are dominated by outliers, the greedy behaviors of these methods easily bias toward outliers, and lead to degraded registration results. An example is shown in Fig. 1(b).

To obtain stronger robustness against outliers, we propose to formulate the registration problem in a novel partial distribution matching (PDM) framework, where we only seek to partially match the distributions. Comparing to the DM based methods, the PDM formulation is more robust against outliers. For example, in Fig. 1(c), the PDM formulation find a more natural solution where only a fraction of points are well matched.

However, existing solutions for PDM problems generally require to compute the correspondence between distributions (Chizat et al., 2018; Chapel et al., 2020), thus they are intractable for registration problems which involve large scale distributions. To handle this issue, we propose an efficient solver for large scale PDM problem. Our method utilizes the partial Wasserstein-1 (PW) discrepancy (Figalli, 2010), which can be efficiently optimized without computing the correspondence. Specifically, we derive the Kantorovich-Rubinstein (KR) duality (Villani, 2003) for the PW discrepancy, and show that its gradient can be explicitly computed via its potential. Based on these results, we propose a partial Wasserstein adversarial network (PWAN), which approximates the potential by a neural network, and the unknown transformation can be trained in an adversarial way with the network. We also incorporate a coherence regularizer for the non-rigid transformation to enforce its smoothness. We note that PWAN generalizes the popular Wasserstein generative adversarial net (WGAN) (Arjovsky et al., 2017) to the PDM problem. An illustration of the proposed PWAN is presented in Fig. 2.

The contribution of this work is summarized as follows.

- Theoretically, we derive the KR duality for the partial mass PW discrepancy, present its differentiability property, and show its gradient can be efficiently computed. We further provide a qualitative description of the KR potentials. More details can be find in Appx. A.3.

- Based on the KR formulation of mass-type (partial mass) PW discrepancy and the closely related distance-type PW discrepancy, we propose a scalable method, called PWAN, for large scale PDM problem. The well-known WGAN is a special case of our method in the DM setting.

- We apply PWAN to point set registration task, where point sets are regarded as discrete distributions. We experimentally show that PWAN exhibits stronger robustness against outliers than the existing methods, and can register point sets accurately even when they are dominated by outliers, such as when they contain a large fraction of noise points, or when they are only partially overlapped.

## 2    RELATED WORK

There is a large body of literatures on point set registration problem (Besl & McKay, 1992; Zhang, 1994; Chui & Rangarajan, 2000; Myronenko & Song, 2010; Maiseli et al., 2017; Vongkulbhisal et al., 2018; Hirose, 2021a). The existing methods can be roughly categorized into two classes, *i.e.*, the

correspondence-based methods and the probabilistic methods. This section only discusses the latter class which is the most related to our method. More discussions can be found in Appx. F.1.

The probabilistic methods solve the registration problem as a DM problem. Specifically, most of these methods smooth the point sets as Gaussian mixture models (GMMs), and then align the distributions via minimizing the robust discrepancies between them. Coherent point drift (CPD) (Myronenko & Song, 2010) and its variants (Hirose, 2021a) are well-known methods is this class, which minimize Kullback-Leibler (KL) divergence between the distributions. Other robust discrepancies, including $L_2$ distance (Ma et al., 2013; Jian & Vemuri, 2011), kernel density estimator (Tsin & Kanade, 2004) and scaled Geman-McClure estimator (Zhou et al., 2016) have also been studied.

The proposed PWAN is related to the existing probabilistic methods because it also regards point sets as distributions. However, it has two major differences from them. First, PWAN directly processes the point sets as un-normalized discrete distributions instead of smoothing them to GMMs, thus it is more concise and natural. Second, PWAN solves a PDM problem instead of the DM problem, as it only requires to match a fraction of distributions, thus it is more robust to outliers.

Some recent works have been devoted to the PDM problem using the Wasserstein-type discrepancy. Bonneel & Coeurjolly (2019) proposed a fast algorithm based on the sliced Wasserstein metric for a special PDM problem, where a small distribution is fully matched to the large one. Various entropy regularized partial or unbalanced discrepancies (Chizat et al., 2018; Séjourné et al., 2019; Chapel et al., 2020) have been utilized for the PDM problem. However, they are generally not scalable to large distributions as they require to compute the correspondence between distributions, or rely on the mini-batch sampling techniques (Fatras et al., 2021). Thus they are not suitable for the registration problem considered in this paper. The distance-type PW discrepancy used in our paper has been utilized for imaging problem (Lellmann et al., 2014; Schmitzer & Wirth, 2019), but these methods do not directly apply to distributions in continuous space. More discussions of the related computational approaches of Wasserstein type discrepancies can be found in Appx. B.

## 3 PRELIMINARIES ON PARTIAL WASSERSTEIN-1 DISCREPANCIES

This section introduces the major tools used in this work, *i.e.*, two types of PW discrepancies between measures: the partial-mass Wasserstein-1 discrepancy (Figalli, 2010; Caffarelli & McCann, 2010) and the bounded-distance Wasserstein-1 discrepancy (Lellmann et al., 2014; Bogachev, 2007). For simplicity, we call them the *mass-type* and the *distance-type* PW discrepancy respectively.

Given two discrete distributions $\alpha = \sum_{x_i \in X} a_i \delta_{x_i}$ and $\beta = \sum_{y_j \in Y} b_j \delta_{y_j}$ on a compact metric space $\Omega \subseteq \mathbb{R}^n$, where $\delta$ is the Dirac function, and $a = [a_i]_{i=1}^q \in \mathbb{R}_+^q$ and $b = [b_j]_{j=1}^r \in \mathbb{R}_+^r$ are the associated mass vectors, the mass-type PW discrepancy (Figalli, 2010) $\mathcal{L}_{M,m}$ is defined as an optimal transport problem which seeks the minimal cost of transporting at least $m$ ($m \leq \min(||a||_1, ||b||_1)$) unit mass from $\alpha$ to $\beta$ where the cost of the transportation equals the distance. Formally, $\mathcal{L}_{M,m}$ is defined as

$$\mathcal{L}_{M,m}(\alpha, \beta) = \min_{\pi \in \Gamma_m(\alpha, \beta)} \sum_{i,j} \pi_{i,j} \, \boldsymbol{d}(x_i, y_j), \tag{1}$$

where $\boldsymbol{d}$ is the metric defined in $\Omega$, $\pi \in \mathbb{R}_+^{q \times r}$ is the transport plan, $\pi_{i,j}$ encodes the amount of mass transported from $x_i$ to $y_j$. The feasible set $\Gamma_m(\alpha, \beta)$ is given by $\Gamma_m(\alpha, \beta) = \{\pi \in \mathbb{R}_+^{q \times r} | \pi \mathbb{1}_r \leq a, \pi^T \mathbb{1}_q \leq b, \mathbb{1}_q^T \pi \mathbb{1}_r \geq m\}$.

The other PW discrepancy used in this work is the distance-type PW discrepancy (Lellmann et al., 2014) $\mathcal{L}_{D,h}$, which is a Lagrangian of $\mathcal{L}_{M,m}$ with the mass constraint softened:

$$\mathcal{L}_{D,h}(\alpha, \beta) = \min_{\pi \in \Gamma_0(\alpha, \beta)} \sum_{i,j} \pi_{i,j} \, (\boldsymbol{d}(x_i, y_j) - h), \tag{2}$$

Note that $\mathcal{L}_{D,h}$ bounds the maximal transport distance by $h$.

In the special complete transport problem where all mass is transported, *i.e.*, when $||a||_1 = ||b||_1$, $m = ||a||_1$ for $\mathcal{L}_{M,m}$ or $h \geq diam(\Omega)$ for $\mathcal{L}_{D,h}$, both $\mathcal{L}_{M,m}$ and $\mathcal{L}_{D,h}$ become equivalent to the well-known Wasserstein-1 distance $\mathcal{W}_1$ (also known as the earth move distance), which, according to

the Kantorovich-Rubinstein duality (Kantorovich, 2006), can be equivalently expressed as

$$\mathcal{W}_1(\alpha, \beta) = \sup_{\boldsymbol{f} \in Lip(\Omega)} \sum_{x_i \in X} a_i \boldsymbol{f}(x_i) - \sum_{y_j \in Y} b_i \boldsymbol{f}(y_j) \tag{3}$$

where $Lip(\Omega)$ represents the Lipschitz-1 function defined on $\Omega$. We called equation (3) the *KR form* of $\mathcal{W}_1$, and $\boldsymbol{f}$ the *potential*. Equation (3) is exploit in WGAN (Arjovsky et al., 2017) to efficiently compute $\mathcal{W}_1$, where $\boldsymbol{f}$ is approximated by a neural network. More detailed preliminaries can be found in Appx. A.

## 4 THE PROPOSED APPROACH

In this section, we present the details of the proposed PWAN. We first formulate the registration problem in Sec. 4.1. Then we present an efficient method to solve the problem in Sec. 4.2 and 4.3. We finally summarize our algorithm in Sec. 4.4.

### 4.1 PROBLEM FORMULATION

Let $Y = \{y_j\}_{j=1}^r \subseteq \Omega$ and $X = \{x_i\}_{i=1}^q \subseteq \Omega$ be the source and reference point sets, where $\Omega \subseteq \mathbb{R}^3$ is a closed bounding box of the points. The corresponding reference and source distributions are $\alpha = \sum_{x_i \in X} a_i \delta_{x_i}$ and $\beta = \sum_{y_j \in Y} b_j \delta_{y_j}$ respectively, where $a_i, b_j \in \mathbf{R}_+$ are the mass assigned to each point and are fixed to be 1 in this work. Let $\mathcal{T}_\theta : \Omega \to \Omega$ denote a differential transformation parametrized by $\theta$ and $\beta_\theta = \sum_{y_j \in Y} b_j \delta_{\mathcal{T}_\theta(y_j)}$ denote the transformed $\beta$. Our goal is to align $\beta_\theta$ to $\alpha$ by solving

$$\min_\theta \mathcal{L}(\alpha, \beta_\theta) + \mathcal{C}(\mathcal{T}_\theta), \tag{4}$$

where discrepancy $\mathcal{L}$ is $\mathcal{L}_{M,m}$ (1) or $\mathcal{L}_{D,h}$ (2), which measures the dissimilarity between $\beta_\theta$ and $\alpha$, and $\mathcal{C}$ is the coherence energy that enforces the spatial smoothness of $\mathcal{T}_\theta$.

### 4.2 REGISTRATION WITH THE PW DISCREPANCIES

In practical registration problems, $\beta_\theta$ and $\alpha$ may contain a large number of outliers or non-overlapping points that should not be matched to the other set. Therefore, in order to avoid biased results, it is important for a registration method to allow the matching of only a fraction of points.

The use of $\mathcal{L}_{M,m}$ and $\mathcal{L}_{D,h}$ in problem (4) naturally leads to desirable partial matchings. To see this, we express them in their respective primal forms, and formulate problem (4) as

$$\sum_{i,j} \pi_{i,j} \boldsymbol{d}(x_i, \mathcal{T}_\theta(y_j)) + \mathcal{C}(\mathcal{T}_\theta) + const, \tag{5}$$

where $const$ is a constant not relevant to $\theta$, and $\pi \in \mathbb{R}^{q \times r}$ represents the correspondence matrix given by the solution of the primal form of $\mathcal{L}_{M,m}(\alpha, \beta_\theta)$ (1) or $\mathcal{L}_{D,h}(\alpha, \beta_\theta)$ (2). A toy example of the correspondence $\pi$ is shown in Fig. 3. As can be seen, $\pi$ only establishes the correspondence between a fraction of points that are close to the other set within the mass threshold $m$ or the distance threshold $h$, while omitting all other points. Therefore, minimizing (5) is simply to align these corresponding point pairs

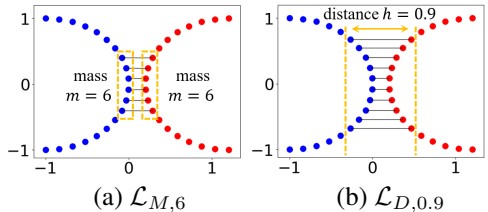

(a) $\mathcal{L}_{M,6}$      (b) $\mathcal{L}_{D,0.9}$

Figure 3: The computed correspondence $\pi$ between $\alpha$ (blue) and $\beta_\theta$ (red).

subjecting to the coherent constraint $\mathcal{C}$. In other words, $\mathcal{L}_{M,m}$ and $\mathcal{L}_{D,h}$ provide two ways to control the ratio of matching based on mass or distance criteria.

Note that according to the Lagrange duality, for each $(\alpha, \beta_\theta, m)$, there exists a $h^*$, such that the solution $\pi$ of $\mathcal{L}_{D,h^*}(\alpha, \beta_\theta)$ recovers to that of $\mathcal{L}_{M,m}(\alpha, \beta_\theta)$. However, the minimization of $\mathcal{L}_{M,m}(\alpha, \beta_\theta)$ in problem (5) is generally not equivalent to that of $\mathcal{L}_{D,h}(\alpha, \beta_\theta)$ with any fixed $h$, because the corresponding $h^*$ depends on $\beta_\theta$, which varies during the optimization process.

Although formulation (5) can indeed handle the partial alignment problem, it is computationally intractable for large scale point sets, because it requires to solve for a matrix $\pi$ of shape $q \times r$ in a linear programing in each iteration. Therefore, a natural question is whether it is possible to avoid the computation of $\pi$ by exploiting the KR duality of $\mathcal{L}_{M,m}$ and $\mathcal{L}_{D,h}$ and re-formulate them in a similar way as $\mathcal{W}_1$ (3).

The answer is affirmative for both $\mathcal{L}_{M,m}$ and $\mathcal{L}_{D,h}$. As for $\mathcal{L}_{D,h}$, its KR form is already known in Bogachev (2007); Lellmann et al. (2014); Schmitzer & Wirth (2019), and we further show that it is valid to compute its gradient under a mild assumption.

**Theorem 1.** $\mathcal{L}_{D,h}(\alpha, \beta)$ *can be equivalently expressed as*

$$\mathcal{L}_{D,h}(\alpha, \beta) = \sup_{\substack{\boldsymbol{f} \in Lip(\Omega) \\ -h \leq \boldsymbol{f} \leq 0}} \sum_{x_i \in X} a_i \boldsymbol{f}(x_i) - \sum_{y_j \in Y} b_j \boldsymbol{f}(y_j) - h m_\beta. \tag{6}$$

*The optimizer of problem* (6) *exists. Let $\boldsymbol{f}$ denote an optimizer of problem* (6)*. When $\mathcal{T}_\theta$ is Lipschitz w.r.t. $\theta$, $\mathcal{L}_{D,h}(\alpha, \beta_\theta)$ is continuous w.r.t. $\theta$, and is differentiable almost everywhere. Furthermore, we have*

$$\nabla_\theta \mathcal{L}_{D,h}(\alpha, \beta_\theta) = -\sum_{y_i \in Y} b_i \nabla_\theta \boldsymbol{f}(\mathcal{T}_\theta(y_i)). \tag{7}$$

As for $\mathcal{L}_{M,m}$, we derive its KR form based on that of $\mathcal{L}_{D,h}$, and show that its gradient can also be computed under a mild assumption.

**Theorem 2.** $\mathcal{L}_{M,m}(\alpha, \beta)$ *can be equivalently expressed as*

$$\mathcal{L}_{M,m}(\alpha, \beta) = \sup_{\substack{\boldsymbol{f} \in Lip(\Omega), h \in \mathbb{R}_+ \\ -h \leq \boldsymbol{f} \leq 0}} \sum_{x_i \in X} a_i \boldsymbol{f}(x_i) - \sum_{y_j \in Y} b_j \boldsymbol{f}(y_j) + h(m - m_\beta). \tag{8}$$

*The optimizer of problem* (8) *exists. Let $(\boldsymbol{f}, h)$ denote an optimizer of problem* (8)*. When $\mathcal{T}_\theta$ is Lipschitz w.r.t. $\theta$, $\mathcal{L}_{M,m}(\alpha, \beta_\theta)$ is continuous w.r.t. $\theta$, and is differentiable almost everywhere. Furthermore, we have*

$$\nabla_\theta \mathcal{L}_{M,m}(\alpha, \beta_\theta) = -\sum_{y_i \in Y} b_i \nabla_\theta \boldsymbol{f}(\mathcal{T}_\theta(y_i)). \tag{9}$$

The proofs of both theorems can be found in Appx. D.3.

With the aid of Theorem 1 and 2, we can optimize $\mathcal{L}_{D,h}$ and $\mathcal{L}_{M,m}$ efficiently. Specifically, we represent the potentials of $\mathcal{L}_{D,h}$ and $\mathcal{L}_{M,m}$ using neural networks $\boldsymbol{f}_{w,h}$ satisfying $-h \leq \boldsymbol{f}_{w,h} \leq 0$ where $h \in \mathbb{R}^+$. To compute $\mathcal{L}_{M,m}(\alpha, \beta_\theta)$, we update $\{w, h\}$ to maximize

$$Loss_{M,m} = \sum_{x_i \in X} a_i \boldsymbol{f}_{w,h}(x_i) - \sum_{y_j \in Y} b_j \boldsymbol{f}_{w,h}(\mathcal{T}_\theta(y_j)) + h(m - m_\beta) - GP(\boldsymbol{f}_{w,h}), \tag{10}$$

and to compute $\mathcal{L}_{D,h}(\alpha, \beta_\theta)$, we update $\{w\}$ to maximize

$$Loss_{D,h} = \sum_{x_i \in X} a_i \boldsymbol{f}_{w,h}(x_i) - \sum_{y_j \in Y} b_j \boldsymbol{f}_{w,h}(\mathcal{T}_\theta(y_j)) - GP(\boldsymbol{f}_{w,h}), \tag{11}$$

where $GP(\boldsymbol{f}) = \kappa \max_{\hat{x} \in X \cup \mathcal{T}_\theta(Y)} \{||\nabla_{\hat{x}} \boldsymbol{f}(\hat{x})||^2, 1\}$ is the gradient penalty (Gulrajani et al., 2017), and $\kappa$ is the strength of the penalty. Then, with the trained network $\boldsymbol{f}_{w,h}$, the gradients $\nabla_\theta \mathcal{L}_{M,m}(\alpha, \beta_\theta)$ and $\nabla_\theta \mathcal{L}_{D,h}(\alpha, \beta_\theta)$ can be computed via back-propagating their respectively losses to $\theta$, thus $\theta$ can be updated via gradient descent.

To show our neural approximations of the PW discrepancies are sufficiently precise, we present a simple example numerically comparing the primal and KR form in Tab. 2 and Fig. 9 in the appendix.

### 4.3   OPTIMIZE THE COHERENCE ENERGY

This section discusses the optimization of the coherence energy, which ensures that the whole $\beta_\theta$ remains spatially smooth in the matching process. First of all, we define the non-rigid transformation $\mathcal{T}_\theta$ parametrized by $\theta = (A, t, V)$ as

$$\mathcal{T}_\theta(y_j) = y_j A + t + V_j, \tag{12}$$

where $y_j \in \mathbb{R}^{1 \times 3}$ represents the coordinate of point $y_j \in Y$, $A \in \mathbb{R}^{3 \times 3}$ is the linear transformation matrix, $t \in \mathbb{R}^{1 \times 3}$ is the translation vector, $V \in \mathbb{R}^{r \times 3}$ is the offset vector of all points in $Y$, and $V_j$ represents the $j$-th row of $V$. Despite its simplicity, $\mathcal{T}_\theta$ includes several useful transformations as its special case. For example, when $V = 0$ and $A \in SO(3)$, $\mathcal{T}_\theta$ becomes the rigid transformation, and when $A = I$ and $t = 0$, $\mathcal{T}_\theta$ becomes the "drift" transformation in Myronenko & Song (2010). Note that $\mathcal{T}_\theta$ is Lipschitz w.r.t. $\theta$ (Proposition 7 in the appendix), thus it satisfies the requirement of Theorem 1 and 2, *i.e.*, it is valid to be used in our registration method.

Now we define the coherence energy similar to Myronenko & Song (2010), *i.e.*, we enforce that $V$ varies smoothly in space. Formally, let $\mathbf{G} \in \mathbb{R}^{r \times r}$ be a kernel matrix, *e.g.*, the Gaussian kernel $\mathbf{G}_\rho(i, j) = e^{-||y_i - y_j||^2/\rho}$, and $\sigma$ be a positive number. The coherence energy is defined as

$$\mathcal{C}(\mathcal{T}_\theta) = \lambda \, Tr(V^T (\sigma \mathcal{I} + \mathbf{G}_\rho)^{-1} V), \tag{13}$$

where $\lambda \in \mathbb{R}_+$ is the strength of constraint, $\mathcal{I}$ is the identity matrix, and $Tr(\cdot)$ is the trace of a matrix.

Since the matrix inversion $(\sigma \mathcal{I} + \mathbf{G}_\rho)^{-1}$ is computationally intractable for large scale point sets, we approximate it via the Nyström method (Williams & Seeger, 2000), and obtain the gradient

$$\frac{\partial \mathcal{C}(\mathcal{T}_\theta)}{\partial V} \approx (2\lambda)(\sigma^{-1} V - \sigma^{-2} \mathbf{Q}(\Lambda^{-1} + \sigma^{-1} \mathbf{Q}^T \mathbf{Q})^{-1} \mathbf{Q}^T V), \tag{14}$$

where $\mathbf{Q} \in \mathbb{R}^{r \times k}$, $\Lambda \in \mathbb{R}^{k \times k}$ is a diagonal matrix, and $k \ll r$. Note the computational cost of (14) is only $O(r)$ if we regard $k$ as a constant. The detailed derivation is presented in Appx. F.3.

## 4.4 THE ALGORITHM AND ANALYSIS

With the methods detailed in Sec. 4.2 and Sec. 4.3, we can finally solve problem (4) efficiently. Formally, we formulate problem (4) as the following mini-max problem

$$\min_\theta \max_{\widetilde{w}} Loss(\alpha, \beta_\theta; \widetilde{w}) + \mathcal{C}(\mathcal{T}_\theta), \tag{15}$$

where $Loss = Loss_{M,m}$ (10) and $\widetilde{w} = \{w, h\}$, or $Loss = Loss_{D,h}$ (11) and $\widetilde{w} = \{w\}$, and we solve it by alternatively updating $\boldsymbol{f}_{w,h}$ and $\mathcal{T}_\theta$.

An illustration of our method is shown in Fig. 2. The detailed algorithm is presented in Alg. 1. For clearness, we refer to PWAN based on $\mathcal{L}_{M,m}$ and $\mathcal{L}_{D,h}$ as mass-type PWAN (m-PWAN) or distance-type PWAN (d-PWAN) respectively.

---

**Algorithm 1** PWAN for point set registration

---

**Input:** reference set $X$, source set $Y$, transform $\mathcal{T}_\theta$, potential network $\boldsymbol{f}_{w,h}$, network update frequency $u$, training type ("*mass*" or "*distance*"), mass threshold $\overline{m}$, distance threshold $\overline{h}$, training step $T$, coherence parameters $(\lambda, \rho, \sigma)$, Nyström parameter $k$.
**Output:** the learned $\theta$.
    **if** training type = "*mass*" **then**
        $\widetilde{w} \leftarrow (w, h);$   $m \leftarrow \overline{m};$   $L \leftarrow Loss_{M,m}$ defined in (10)
    **else if** training type = "*distance*" **then**
        $\widetilde{w} \leftarrow (w);$   $h \leftarrow \overline{h};$   $L \leftarrow Loss_{D,h}$ defined in (11)
    **end if**
    **for** $t = 1, \ldots, T$ **do**
        Obtain the transformed distribution $\beta_\theta$.
        **for** $= 1, \ldots, u$ **do**
            Compute $\partial L/\partial \widetilde{w}$ by back-propagating $L$ through $\widetilde{w}$.
            Update $\widetilde{w}$ by ascending the gradient $\partial L/\partial \widetilde{w}$.           ▷ Potential learning
        **end for**
        Compute $\partial \mathcal{C}/\partial \theta$ using $(\lambda, \rho, \sigma, k)$ according to (14).
        Compute $\partial L/\partial \theta$ by back-propagating $L$ through $\theta$.
        Update $\theta$ by descending the gradient $\partial(L + \mathcal{C})/\partial \theta$.          ▷ Registration
    **end for**

---

To provide an intuitive explanation why solving problem (15) can lead to partial alignment, we first visualize the learned potential on a toy example in Fig. 4. As can be seen, the potential of PWAN

attains the upper or lower bound (0 or $-h$) in some regions, thus the gradient within these regions is strictly 0, *i.e.*, the potential is strictly "flat" in these regions. This observation is formally stated and proved in Proposition 11 and 12 in the appendix.

Due to the existence of flat regions, PWAN can automatically discard a fraction of points during the registration process. Specifically, if we omit the coherent energy, in each iteration of Alg. 1, PWAN moves $\beta_\theta$ along the gradient of potential. Therefore, PWAN only moves the fraction of points with non-zero gradient, while leaving the points with zero gradients (in flat regions) fixed. For the case in Fig. 4, only the leftmost 3 points in $\beta_\theta$ will be moved leftward in current iteration, while others will stay fixed. In other words, PWAN seeks to match a fraction of points instead of the whole point sets.

More discussions can be found in Appx. F.4.

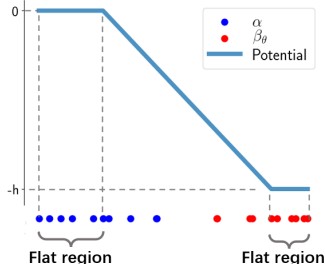

Figure 4: The learned potential on toy 1-dimensional point sets.

## 5 EXPERIMENTS AND ANALYSIS

In this section, we experimentally evaluate the proposed PWAN on point set registration tasks. After describing the experiment settings in Sec. 5.1, we first present a toy example to highlight the robustness of the PW discrepancies in Sec. 5.2. Then we compare PWAN against the state-of-the-art methods in Sec. 5.3, and discuss its scalability in Sec. 5.4. We finally evaluate PWAN on two real datasets in Sec. 5.5. More experimental results are given in Appx. F.

### 5.1 EXPERIMENT SETTINGS

As shown in Fig. 5, we use three synthesized datasets in our experiments: bunny, armadillo and monkey. The bunny and armadillo datasets are from the Stanford Repository (Standford), and the monkey dataset is from the internet. The number of points in these shape are $8, 171$, $106, 289$ and $7, 958$ respectively. Following Hirose (2021b), we artificially deform these sets, and we evaluate the registration results via the mean square error (MSE).

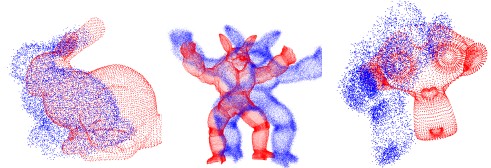

Figure 5: The synthesized datasets used in our experiments. The source point sets (blue) are synthesized by bending the reference point sets (red) in a non-linear manner.

We use a 5-layer point-wise multi-layer perception as our network (Fig. 10 in the appendix), and the parameters used in the experiments are given in Appx. F.5.

### 5.2 COMPARISON OF SEVERAL DISCREPANCIES

To provide an intuition of our PDM formulation for registration, we compare the PW discrepancy with two representative robust discrepancies used in DM based registration methods, *i.e.*, KL divergence (Hirose, 2021a) and $L_2$ distance (Jian & Vemuri, 2011), on a toy 1-dimensional example. For now, we do not consider the coherence energy.

We construct the toy point sets $X$ and $Y$ shown in Fig. 6(a), where we first sample 10 equi-spaced data points in interval $[0, 3]$, then we define $Y$ by translating the data points by a distance $t$, and define $X$ by adding $N$ outliers in a narrow interval $[7.8, 8.2]$ to the data points. For $N = \{1, 10, 10^3\}$, we record four discrepancies: KL, $L_2$, $\mathcal{L}_{M,10}$ and $\mathcal{L}_{D,2}$ between $X$ and $Y$ as a function of $t$, and present the results from Fig. 6(b) to Fig. 6(e).

As can be seen, there exist two alignment modes in this experiment, *i.e.*, $t = 0$ and $t = 6.5$, which respectively correspond to the correct alignment and the degraded alignment where $Y$ is matched to outliers. When the number of outliers is small, *e.g.*, $N = 1$, all discrepancies admit a deep local minimum $t = 0$. However, as for KL and $L_2$ divergence, the local minimum $t = 0$ gradually vanishes and they gradually bias toward $t = 6.5$ as $N$ increases. In contrast, $\mathcal{L}_{M,10}$ and $\mathcal{L}_{D,2}$ do not suffer

from this issue, *i.e.*, the local minimum $t = 0$ remains deep regardless of the value of $N$, and the local minimum $t = 6.5$ is always shallower than the local minimum $t = 0$.

The results show a key advantage of PDM against DM for registration: When the number of remote outliers is large, the DM formulations (KL and $L_2$) always tend to converge to the biased result, while the correct alignment of PDM formulation is not influenced by the remote outliers, and it is less likely to converge to the biased result.

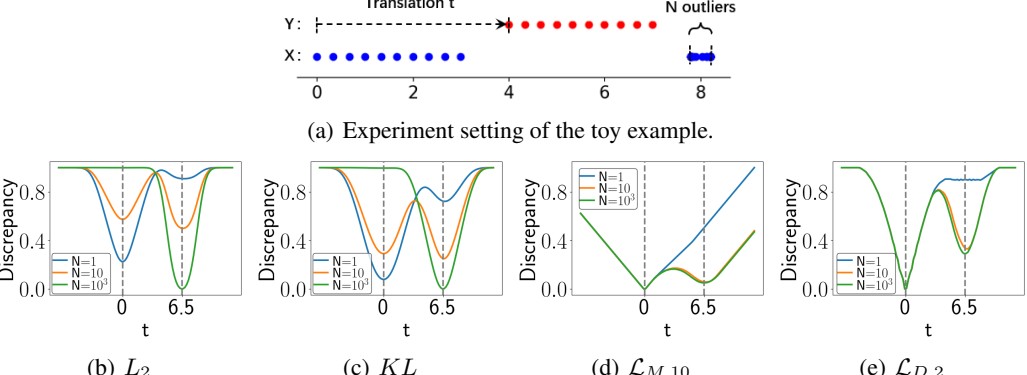

(a) Experiment setting of the toy example.

(b) $L_2$    (c) $KL$    (d) $\mathcal{L}_{M,10}$    (e) $\mathcal{L}_{D,2}$

Figure 6: Comparison of different discrepancies on a toy point set.

## 5.3 Evaluation of the Registration Accuracy

We compare the performance of PWAN with four state-of-the-art methods: CPD (Myronenko & Song, 2010), GMM-REG (Jian & Vemuri, 2011), BCPD (Hirose, 2021a) and TPS-RPM (Chui & Rangarajan, 2000). We evaluate them on the following two artificial datasets.

**Point sets with extra noise points.** We first sample $N = 500$ random points from each of the original and the deformed sets as the source and reference sets respectively. Then we add extra uniformly distributed noise points to the reference set, and we normalize both sets to mean 0 and variance 1. We vary the number of outliers from 100 to 600 at an interval of 100, *i.e.*, the outlier/non-outlier ratio varies from 0.2 to 1.2 at an interval of 0.2.

**Partially overlapped point sets.** We first sample $N = 1000$ random points from each of the original and the deformed sets as the source and reference sets respectively. Then for each set, we intersect it with a random plane, and we retain the points in one side of the plane and discard all points in the other side. We vary the retain ratio $s$ from 0.7 to 1.0 at an interval of 0.05 for both the source and the reference sets, thus the minimal ratios of overlapping area are $(2s - 1)/s = 0.57, 0.67, 0.75, 0.82, 0.89, 0.94$ and 1 accordingly, and the minimal corresponding mass is $\underline{m} = (2s - 1) * 1000$.

We evaluate m-PWAN with $m = 500$ (equivalently d-PWAN with $h = +\infty$) in the first experiment, while we evaluate m-PWAN with $m = \underline{m}$ and d-PWAN with $h = 0.05$ in the second experiment. We run all methods 100 times and report median and standard deviation of MSE in Fig. 7.

Fig. 7(a) presents the results of the first experiment. The median registration error of PWAN is generally comparable with TPS-RPM, and is much lower than the other methods. In addition, the standard deviations of the results of PWAN are much lower than that of TPS-RPM. This suggests that PWAN are more robust against outliers than baseline methods. Fig. 7(b) presents the results of the second experiment. Two types of PWANs perform comparably, and they outperform the baseline methods by a large margin in terms of both median and standard deviations when the overlap ratio is low, while perform comparably with them when the data is fully overlapped. This result suggests the proposed PWAN can effectively register the partially overlapped sets.

## 5.4 Evaluation of the efficiency

To evaluate the efficiency of PWAN, we first need to investigate the influence of its parameters. In particular, the most important parameter that affects the efficiency is the network update frequency $u$, which controls the tradeoff between the efficiency and effectiveness. Specifically, larger $u$ leads to more accurate estimation of the potential and the gradient, while smaller $u$ allows for faster estimations. We quantify the influence of $u$ using the bunny dataset and report the results in the left

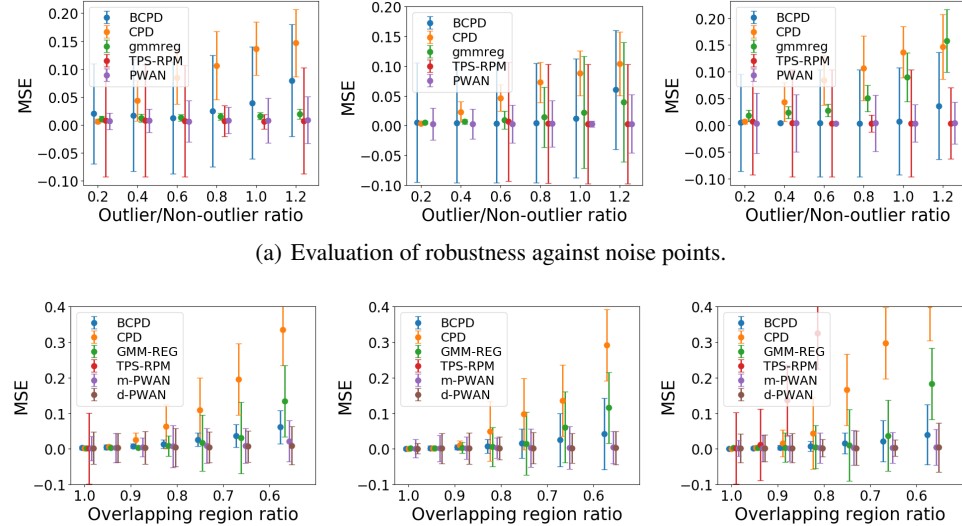

(a) Evaluation of robustness against noise points.

(b) Evaluation of robustness against partial overlapping.

Figure 7: Registration accuracy of the bunny (left), monkey (middle) and armadillo (right) datasets. The error bars represent the medians and standard deviations of MSE.

panel of Fig. 8. As can be seen, both MSE and the variance of MSE decrease as $u$ increases, while the computation time increases proportionally with $u$. This result suggests that more accurate and stable results can be achieved at the expense of higher computational cost, *i.e.*, larger $u$.

We benchmark the computation time of different methods on a computer with two Nvidia GTX TITAN GPUs and an Intel i7 CPU. We fix $u = 20$ for PWAN. We sample $q = r$ points from the bunny shape, where $q$ varies from $10^3$ to $7 \times 10^5$. PWAN is run on the GPU while the other methods are run on the CPU. We also implement a multi-GPU version of PWAN where the potential network is updated in parallel. We

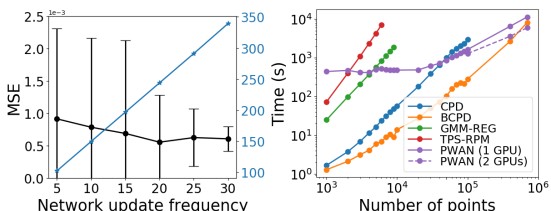

Figure 8: Scalability of our method.

run each method 10 times and report the mean of the computation time in the right panel of Fig. 8. As can be seen, BCPD is the fastest method when $q$ is small, and PWAN is comparable with BCPD when $q$ is near $10^6$. In addition, the 2-GPU version PWAN is faster than the single GPU version, and it is faster than BCPD when $q$ is larger than $5 \times 10^5$.

## 5.5 EVALUATION ON REAL DATA

To demonstrate the capability of PWAN on handling point sets with non-artificial deformations, we evaluate it on a human face dataset (Zhang et al., 2008) and a 3D human dataset (DataSet). The details of this experiment are presented in Appx. F.9 and Appx. F.10.

## 6 CONCLUSION

In this paper, we formulate the point set registration task as a PDM problem, where point sets are regarded as discrete distributions and are only required to be partially matched. In order to solve the PDM problem efficiently, we derived the KR form and the gradient for the PW discrepancy. Based on the theory, we proposed the PWAN method for PDM problem, and applied it to practical point sets registration task. We experimentally show that PWAN can effectively handle the point sets dominated by outliers, including those containing large fraction of noise or being partially overlapped.

There are several issues need further study. First, the computation time of PWAN is still relatively high. A possible approach to accelerate PWAN is to incorporate a forward generator network as in Sarode et al. (2019) or to use the discriminative training as in Vongkulbhisal et al. (2018). Second, it is interesting to explore PWAN in other PDM problems, such as domain adaption (Hu et al., 2020), pose estimation (Kuhnke & Ostermann, 2019) and multi-label learning (Yan & Guo, 2019).

ACKNOWLEDGMENTS

This work was supported by the National Natural Science Foundation of China under Grant 61922065, Grant 62101390, Grant 41820104006, the Fundamental Research Funds for the Central Universities under Grant 2042021kf0038, and the National Post-Doctoral Program for Innovative Talents under Grant BX20200248.

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

# APPENDIX

In this appendix, we provide theoretical justifications of our algorithm and more experimental results.

We first present a more general introduction of the optimal transport problem and fix the notations in Sec. A. Then we introduce existing computational approaches of optimal transport problem in Sec. B. We derive our formulation in Sec. C and the gradient in Sec. D. We further discuss the properties of our formulations in Sec. E. Finally, we present more details for the main text in Sec F.

## A  PRELIMINARIES

This section introduces the optimal transport (OT) problem and fix the notations for later sections.

### A.1  OPTIMAL TRANSPORT

OT is a classic problem dating back to Monge and Kantorovich. It requires to solve the following transportation problem: let $\alpha$ be the distribution of warehouses storing some raw materials, and $\beta$ be the distribution of factories requiring these materials. Assuming the total mass of materials stored in the warehouse is $m_\alpha$, and the total mass of materials required by the factories is $m_\beta$. How to transport at least $m$ ($m \leq \min(m_\alpha, m_\beta)$) mass materials from $\alpha$ to $\beta$ so that the total cost is minimized?

Formally, let $\Omega$ be compact metric space and $\mathcal{M}_+(\Omega)$ be the set of non-negative measures defined on $\Omega$. Define the source and target measures $\alpha, \beta \in \mathcal{M}_+(\Omega)$, and a continuous cost function $c : \Omega \times \Omega \to \mathbb{R}^+$. OT seeks to solve the following optimization problem

$$\mathcal{L}_{M,m}(\alpha, \beta) = \inf_{\pi \in \Gamma_m(\alpha, \beta)} \int_{\Omega \times \Omega} c(x, y) d\pi(x, y), \tag{16}$$

where $\Gamma_m(\alpha, \beta)$ are the set of non-negative measures $\pi$ defined on $\Omega \times \Omega$ satisfying

$$\pi(A \times \Omega) \leq \alpha(A), \quad \pi(\Omega \times A) \leq \beta(A) \quad and \quad \pi(\Omega \times \Omega) \geq m$$

for all measurable set $A \subseteq \Omega$. For ease of notations, we abbreviate $m_\alpha = \alpha(\Omega)$, $m_\beta = \beta(\Omega)$ and $m(\pi) = \pi(\Omega \times \Omega)$.

In the complete OT problem, it is generally assumed that $m_\alpha = m_\beta = m$, *i.e.*, the source and the target distributions contain equal mass of materials, and *all* materials should be transported. However, the more general partial optimal transport (POT) problem (Figalli, 2010; Caffarelli & McCann, 2010) only requires $m \leq min(m_\alpha, m_\beta)$, *i.e.*, the source and target distributions may contain different mass of materials, and only a *partial* of mass is required to transported.

### A.2  PARTIAL WASSERSTEIN-1 DISCREPANCY

In this paper, we focus on a specific POT problem where the cost function is a distance in the metric space $\Omega$, *i.e.*, $c(x, y) = d(x, y)$, where $d$ is the distance function defined in $\Omega$. In this case, we called $\mathcal{L}_{M,m}$ the mass-type partial Wasserstein-1 (PW) discrepancy.

In the complete OT case, this type of OT problem defines the Wasserstein-1 metric, which is also known as the earth move distance (EMD) between distributions. According to the Kantorovich-Rubinstein duality (Kantorovich, 2006), the Wasserstein-1 metric can be equivalently expressed as

$$\mathcal{W}_1(\alpha, \beta) = \sup_{\boldsymbol{f} \in Lip(\Omega)} \int_\Omega \boldsymbol{f} d\alpha - \int \boldsymbol{f} d\beta. \tag{17}$$

Note that this formulation offers a more efficient implement of Wasserstein-1 metric than the primal form (16), as it only requires to solve for a function with a local constraint in $\Omega$ instead of a transport plan with global constraints in $\Omega \times \Omega$.

Several methods have been proposed to generalize (17) to the unbalanced OT (Chizat et al., 2018; Schmitzer, 2019), *i.e.*, OT problems with extra regularizers. See Schmitzer & Wirth (2019) for an unified framework of this type of generalizations. Amongst these generalizations, KR metric (Lellmann et al., 2014) is closely related to the POT problem (16) considered in this paper, as it can be regarded

as a Lagrangian of problem (16) where the mass constraint is soften. Specifically, the primal form KR metric with parameter $h > 0$ is defined as

$$KR_h(\alpha, \beta) = \inf_{\pi \in \Gamma_0(\alpha, \beta)} \int_{\Omega \times \Omega} d(x, y) d\pi(x, y) - hm(\pi) + \frac{h}{2}(m_\alpha + m_\beta). \tag{18}$$

It is important to notice that the KR metric has a natural explanation that it requires to find a optimal plan whose *transport distance does not exceed* $h$. This can be seen by re-writing problem (18) as $KR_h(\alpha, \beta) = \inf \int (d(x, y) - h) d\pi(x, y) + const$, and noticing that if $d(x, y) > h$, then the solution $\pi^*$ to problem (18) should satisfy $\pi^*(x, y) = 0$.

Importantly, the KR metric is known to have an equivalent form (Lellmann et al., 2014; Schmitzer & Wirth, 2019)

$$KR_h(\alpha, \beta) = \sup_{\substack{\boldsymbol{f} \in Lip(\Omega) \\ |\boldsymbol{f}| \leq \frac{h}{2}}} \int_\Omega \boldsymbol{f} d\alpha - \int \boldsymbol{f} d\beta, \tag{19}$$

which is very similar to (17). We called formulations (17) and (19) *KR form*s, and the solution to KR forms *potential*s.

## A.3 THEORETICAL CONTRIBUTIONS

The main theoretical contributions of this work are summarized as follows.

- We present the KR form of $\mathcal{L}_{M,m}$ in Proposition 3 in Sec. C.
- We prove the differentiability of the KR form of $\mathcal{L}_{M,m}$ and derive its gradient in Sec. D.
- We characterize of the potential of the KR form of $\mathcal{L}_{M,m}$ in Sec. E. The main result is Proposition 11.

## A.4 NOTATIONS

- $(\Omega, d)$: a metric $d$ associated to a compact metric space $\Omega$. For example, $\Omega$ can be a closed cubic in $\mathbb{R}^3$ and $d$ can be the Euclidean distance.
- $C(\Omega)$: the set of continuous bounded function defined on $\Omega$ equipped with the supreme norm.
- $Lip(\Omega) \subseteq C(\Omega)$: the set of 1-Lipschitz function defined on $\Omega$.
- $\mathcal{M}(X)$: the space of Radon measures on space $X$.
- $\pi_\#^1$: The marginal of $\pi$ on its first variable. Similarly, $\pi_\#^2$ represents the marginal of $\pi$ on its second variable.
- Given a function $\mathbf{F}$: $X \to \mathbb{R} \cup +\infty$, the Fenchel conjugate of $\mathbf{F}$ is denoted as $\mathbf{F}^*$ and is given by:

$$\mathbf{F}^*(x^*) = \sup_{x \in X} <x, x^*> -\mathbf{F}(x), \quad \forall x^* \in X^* \tag{20}$$

where $X^*$ is the dual space of $X$ and $< \cdot >$ is the dual pairing.

## B EXISTING COMPUTATIONAL APPROACHES OF OT PROBLEM

The computation of OT problem is an active field in machine learning. In this section, we briefly discuss three major classes of approaches and relate our method to the existing ones.

One class of the most well-developed OT solver is based on the entropic regularizer. This class of approaches relax the primal problem by adding an entropy regularizer to the transport plan, then solve the relaxed problem via the Sinkhorn algorithm (Cuturi, 2013). However, the direct application of the Sinkhorn algorithm has two drawbacks. First, the entropic bias introduced by the regularizer always leads to undesired behaviors. Second, the computational cost is high for large scale problems, since the Sinkhorn algorithm iteratively updates the whole transport matrix. Some works have been devoted to address these two issues. To get rid of the bias, Genevay et al. (2018; 2019) proposed the Sinkhorn divergence as an unbiased version of the entropic OT. To improve the efficiency, Schmitzer (2019) proposed some acceleration techniques such as muti-scale computing, and Fatras et al. (2020)

proposed to consider the mini-batch OT problem to avoid the computation of the complete transport plan. The generalization of this type of approaches to unbalanced or partial OT problem was studied in Benamou et al. (2015); Chizat et al. (2018); Séjourné et al. (2019); Fatras et al. (2021).

The second class of approaches solve the sliced OT problem (Bonneel et al., 2015; Kolouri et al., 2016). They avoid computing OT problems in high dimensional space by projecting the distributions onto random 1-dimensional lines, and solving a 1-dimensional OT problem each time. Due to their simplicity and efficiency, this class of approaches have been applied to several fields in computer vision, such as generative modelling (Deshpande et al., 2018) and texture synthesis (Heitz et al., 2021). Recently, Bonneel & Coeurjolly (2019) proposed an algorithm for a special case of partial sliced OT problem, where a small distribution is completely matched to a fraction of a large distribution. However, this method does not handle the general partial sliced OT problem.

Our method belongs to the third class of approaches which focus on a specific type of OT problem: the Wasserstein-1 type problem. The foundation of this type of approach is the Kantorovich-Rubinstein duality (17), which allows to efficiently compute Wasserstein-1 distance by learning a Lipschitz function. This property was directly exploited in the popular Wasserstein GAN model (Arjovsky et al., 2017). Some works (Lellmann et al., 2014; Schmitzer & Wirth, 2019) generalized this duality to the unbalanced Wasserstein-1 type problem and applied them to imaging problems, but the exact partial Wasserstein-1 problem ((16) with $c = d$) has not been considered. Our method completes these approaches in a sense that we solve the exact partial Wasserstein-1 problem. Besides, unlike the existing works (Lellmann et al., 2014; Schmitzer & Wirth, 2019) which only handle the distributions in discrete image space, our method handles distributions in the continuous space, thus it is more suitable for applications in machine learning such as point sets registration.

## C OUR FORMULATIONS

In this section, we derive the KR formulation of $\mathcal{L}_{M,m}$. The main result is Theorem 3.

First of all, we derive the Fenchel-Rockafellar dual of $\mathcal{L}_{M,m}(\alpha, \beta)$.

**Proposition 1** (Dual form of $\mathcal{L}_{M,m}$). *Problem* (16) *can be equivalently expressed as*

$$\mathcal{L}_{M,m}(\alpha, \beta) = \sup_{(\boldsymbol{f},\boldsymbol{g},h)\in \mathbf{R}} \int_{\Omega} \boldsymbol{f} d\alpha + \int_{\Omega} \boldsymbol{g} d\beta + mh. \tag{21}$$

*where the feasible set* $\mathbf{R}$ *is*

$$\mathbf{R} = \Big\{ (\boldsymbol{f},\boldsymbol{g},h) \in C(\Omega) \times C(\Omega) \times \mathbb{R}_+ | \boldsymbol{f} \leq 0, \ \boldsymbol{g} \leq 0, \ c(x,y) - h - \boldsymbol{f}(x) - \boldsymbol{g}(y) \geq 0, \forall x, y \in \Omega \Big\} \tag{22}$$

*In addition, the infimum in Problem* (16) *is attained.*

*Proof.* We prove this proposition via Fenchel-Rockafellar duality. We first define space $E$: $C(\Omega) \times C(\Omega) \times \mathbb{R}$, space $F$: $C(\Omega \times \Omega)$, and a linear operator $\mathcal{A}$: $E \to F$ as

$$\mathcal{A}(\boldsymbol{f},\boldsymbol{g},h) : (x,y,h) \to \boldsymbol{f}(x) + \boldsymbol{g}(y) + h; \quad \forall \boldsymbol{f}, \boldsymbol{g} \in C(\Omega), \ \forall h \in \mathbb{R}, \ \forall x, y \in \Omega. \tag{23}$$

Then we introduce a convex function $\mathbf{H}$: $F \to \mathbb{R} \cup +\infty$ as

$$\mathbf{H}(u) = \begin{cases} 0 & if \ u \geq -c \\ +\infty & else \end{cases} \tag{24}$$

and $\mathbf{L}$: $E \to \mathbb{R} \cup +\infty$ as

$$\mathbf{L}(\boldsymbol{f},\boldsymbol{g},h) = \begin{cases} \int \boldsymbol{f} d\alpha + \int \boldsymbol{g} d\beta + hm & if \ \boldsymbol{f} \geq 0, \ \boldsymbol{g} \geq 0, \ h \leq 0 \\ +\infty & else \end{cases} \tag{25}$$

We can check when $\boldsymbol{f} \equiv \boldsymbol{g} \equiv 1$ and $h = -1$, $\mathbf{H}$ is continuous at $\mathcal{A}(\boldsymbol{f},\boldsymbol{g},h)$. Thus by Fenchel-Rockafellar duality, we have

$$\inf_{(\boldsymbol{f},\boldsymbol{g},h)\in E} \mathbf{H}(\mathcal{A}(\boldsymbol{f},\boldsymbol{g},h)) + \mathbf{L}(\boldsymbol{f},\boldsymbol{g},h) = \sup_{\pi \in \mathcal{M}(\Omega \times \Omega)} -\mathbf{H}^*(-\pi) - \mathbf{L}^*(\mathcal{A}^*\pi) \tag{26}$$

We first compute the Fenchel dual $\mathbf{H}^*(-\pi)$ and $\mathbf{L}^*(\mathcal{A}^*\pi)$ in the right-hand side of (26). For arbitrary $\pi \in \mathcal{M}(\Omega \times \Omega)$, we have

$$\mathbf{H}^*(-\pi)$$
$$= \sup_{u \in F}\left\{\int(-u)d\pi - \mathbf{H}(u)\right\}$$
$$= \sup_{u \in F}\left\{\int(-u)d\pi \,\middle|\, u(x,y) \geq -c(x,y),\ \forall(x,y) \in \Omega \times \Omega\right\}$$
$$= \sup_{u \in F}\left\{\int u d\pi \,\middle|\, u(x,y) \leq c(x,y),\ \forall(x,y) \in \Omega \times \Omega\right\}$$

It is easy to see that if $\pi$ is a non-negative measure, then this supremum is $\int c d\pi$, otherwise it is $+\infty$. Thus

$$\mathbf{H}^*(-\pi) = \begin{cases} \int c(x,y)d\pi(x,y) & if\ \pi \in \mathcal{M}_+(\Omega \times \Omega) \\ +\infty & else \end{cases} \tag{27}$$

Similarly, we have

$$\mathbf{L}^*(\mathcal{A}^*\pi)$$
$$= \sup_{(\boldsymbol{f},\boldsymbol{g},h) \in E}\left\{<(\boldsymbol{f},\boldsymbol{g},h), A^*\pi> -\mathbf{L}(\boldsymbol{f},\boldsymbol{g},h)\right\}$$
$$= \sup_{(\boldsymbol{f},\boldsymbol{g},h) \in E}\left\{<A(\boldsymbol{f},\boldsymbol{g},h),\pi> -(\int \boldsymbol{f}d\alpha + \int \boldsymbol{g}d\beta + hm)|\boldsymbol{f},\boldsymbol{g} \geq 0,\ h \leq 0\right\}$$
$$= \sup_{(\boldsymbol{f},\boldsymbol{g},h) \in E}\left\{\int \boldsymbol{f}d(\pi_{\#}^1 - \alpha) + \int \boldsymbol{g}d(\pi_{\#}^2 - \beta) + h(\pi(\Omega \times \Omega) - m)|\boldsymbol{f},\boldsymbol{g} \geq 0,\ h \leq 0\right\}$$

If $(\alpha - \pi_{\#}^1)$ and $(\beta - \pi_{\#}^2)$ are non-negative measures, and $\pi(\Omega \times \Omega) - m \geq 0$, this supremum is 0, otherwise it is $+\infty$. Thus

$$\mathbf{L}^*(\mathcal{A}^*\pi) = \begin{cases} 0 & if(\alpha - \pi_{\#}^1) \in \mathcal{M}_+(\Omega),\ (\beta - \pi_{\#}^2) \in \mathcal{M}_+(\Omega),\ \pi(\Omega \times \Omega) \geq m \\ +\infty & else \end{cases} \tag{28}$$

In addition, the left-hand side of (26) reads

$$\inf_{(\mathbf{f},\mathbf{g},\mathbf{h}) \in E} \mathbf{H}(\mathcal{A}(\mathbf{f},\mathbf{g},\mathbf{h})) + \mathbf{L}(\boldsymbol{f},\boldsymbol{g},h)$$
$$= \inf_{(\boldsymbol{f},\boldsymbol{g},h) \in E}\left\{\int \boldsymbol{f}d\alpha + \int \boldsymbol{g}d\beta + hm|\ \boldsymbol{f},\boldsymbol{g} \geq 0,\ h \leq 0,\ \boldsymbol{f}(x) + \boldsymbol{g}(y) + h \geq -c(x,y), \forall x,y \in \Omega\right\}$$
$$= -\sup_{(\boldsymbol{f},\boldsymbol{g},h) \in E}\left\{\int \boldsymbol{f}d\alpha + \int \boldsymbol{g}d\beta + hm|\ \boldsymbol{f},\boldsymbol{g} \leq 0,\ h \geq 0,\ \boldsymbol{f}(x) + \boldsymbol{g}(y) + h \leq c(x,y), \forall x,y \in \Omega\right\}$$

Finally, by inserting these terms into (26), we have

$$\sup_{\substack{(\boldsymbol{f},\boldsymbol{g},h) \in E \\ \boldsymbol{f},\boldsymbol{g} \leq 0, h \geq 0 \\ \boldsymbol{f}(x) + \boldsymbol{g}(y) + h \leq c(x,y)\forall x,y \in \Omega}} \int \boldsymbol{f}d\alpha + \int \boldsymbol{g}d\beta + hm = \inf_{\substack{\pi \in \mathcal{M}_+(\Omega \times \Omega) \\ (\alpha-\pi_{\#}^1) \in \mathcal{M}_+(\Omega),(\beta-\pi_{\#}^2) \in \mathcal{M}_+(\Omega) \\ \pi(\Omega \times \Omega) \geq m}} \int c(x,y)d\pi(x,y),$$

which proves (21).

In addition, we can also check right-hand side of (26) is finite, since we can always construct independent coupling $\widetilde{\pi} = \frac{\sqrt{m}}{\alpha(\Omega)}\alpha \otimes \frac{\sqrt{m}}{\beta(\Omega)}\beta$, such that $\widetilde{\pi} \in \mathcal{M}_+(\Omega \times \Omega)$, $\widetilde{\pi}_{\#}^1 = \frac{m}{\alpha(\Omega)}\alpha \leq \alpha$, $\widetilde{\pi}_{\#}^2 = \frac{m}{\beta(\Omega)}\beta \leq \beta$ and $\widetilde{\pi}(\Omega \times \Omega) = m$. Thus the Fenchel-Rockafellar duality suggests the infimum is attained. $\qquad\square$

Then we define the following Lagrangian POT problem:

$$\mathcal{L}_{D,h}(\alpha,\beta) = \inf_{\pi \in \Gamma_0(\alpha,\beta)} \int_{\Omega \times \Omega} c(x,y)d\pi(x,y) - hm(\pi). \tag{29}$$

where $h > 0$ is the Lagrange multiplier. When $c(x, y) = d(x, y)$, we immediately obtain the KR form of this problem by reformulating the KR metric (18):

$$\mathcal{L}_{D,h}(\alpha, \beta) = \sup_{\substack{\boldsymbol{f} \in Lip(\Omega) \\ -h \leq \boldsymbol{f} \leq 0}} \int_\Omega \boldsymbol{f} d\alpha - \int_\Omega \boldsymbol{f} d\beta - h m_\beta. \tag{30}$$

Similar to Proposition 1, we derive the Fenchel-Rockafellar dual form of $\mathcal{L}_{D,h}$.

**Proposition 2** (Dual form of $\mathcal{L}_{D,h}$). *Problem* (29) *can be equivalently expressed as*

$$\mathcal{L}_{D,h}(\alpha, \beta) = \sup_{(\boldsymbol{f}, \boldsymbol{g}) \in \mathbf{R}(h)} \int_\Omega \boldsymbol{f} d\alpha + \int_\Omega \boldsymbol{g} d\beta. \tag{31}$$

*where the feasible set is*

$$\mathbf{R}(h) = \Big\{ (\boldsymbol{f}, \boldsymbol{g}) \in C(\Omega) \times C(\Omega) | \boldsymbol{f} \leq 0, \ \boldsymbol{g} \leq 0, \ c(x, y) - h - \boldsymbol{f}(x) - \boldsymbol{g}(y) \geq 0, \forall x, y \in \Omega \Big\} \tag{32}$$

*In addition, the infimum in Problem* (29) *is attained.*

*Proof.* This proposition can be proved in similarly to Proposition 1, so we omit the proof here. $\square$

Now, by comparing Proposition 2 with Proposition 1, we can see that $\mathcal{L}_{D,h}$ and $\mathcal{L}_{M,m}$ are related by

$$\begin{aligned} \mathcal{L}_{M,m} &= \sup_{(\boldsymbol{f}, \boldsymbol{g}, h) \in \mathbf{R}} \int_\Omega \boldsymbol{f} d\alpha + \int_\Omega \boldsymbol{g} d\beta + mh \\ &= \sup_{h \in \mathbb{R}_+} \mathcal{L}_{D,h} + mh. \end{aligned} \tag{33}$$

Therefore, we obtain the KR form of $\mathcal{L}_{M,m}$ by inserting (30) into (33).

**Theorem 3** (KR form of $\mathcal{L}_{M,m}$). *When $c(x, y) = d(x, y)$, problem* (16) *can be reformulated as*

$$\mathcal{L}_{M,m}(\alpha, \beta) = \sup_{\substack{\boldsymbol{f} \in Lip(\Omega), h \in \mathbf{R}_+ \\ -h \leq \boldsymbol{f} \leq 0}} \int_\Omega \boldsymbol{f} d\alpha - \int_\Omega \boldsymbol{f} d\beta + h(m - m_\beta) \tag{34}$$

*or equivalently as*

$$\mathcal{L}_{M,m}(\alpha, \beta) = \sup_{\boldsymbol{f} \in Lip(\Omega), \boldsymbol{f} \leq 0} \int_\Omega \boldsymbol{f} d\alpha - \int_\Omega \boldsymbol{f} d\beta - \inf(\boldsymbol{f})(m - m_\beta). \tag{35}$$

*Proof.* We obtain equation (34) by inserting (30) into (33) and merging two supremums. As for equation (35), note that given a fixed $\boldsymbol{f} \in C(\Omega)$ in problem (34), the optimal $h$ is simply $-\inf(\boldsymbol{f}) < +\infty$. So we can replace $h$ by $-\inf(\boldsymbol{f})$ in problem (34) to obtain problem (35). $\square$

For clearness, we define some functionals associated to $\mathcal{L}_{M,m}$ and $\mathcal{L}_{D,h}$.

**Definition 1** ($\mathbf{L}_{M,m}$ and $\overline{\mathbf{L}_{M,m}}$). *Define functional $\mathbf{L}_{M,m}$ associated to problem* (34) *as*

$$\mathbf{L}_{M,m}^{\alpha, \beta}(\boldsymbol{f}, h) = \begin{cases} \int_\Omega \boldsymbol{f} d(\alpha - \beta) + h(m - m_\beta) & h \in \mathbb{R}_+ , \ \boldsymbol{f} \in Lip(\Omega) , \ -h \leq \boldsymbol{f} \leq 0 \\ -\infty & else, \end{cases} \tag{36}$$

*and functional $\overline{\mathbf{L}_{M,m}}$ associated to problem* (35) *as*

$$\overline{\mathbf{L}_{M,m}^{\alpha, \beta}}(\boldsymbol{f}) = \begin{cases} \int_\Omega \boldsymbol{f} d(\alpha - \beta) - \inf(\boldsymbol{f})(m - m_\beta) & \boldsymbol{f} \in Lip(\Omega) , \ \boldsymbol{f} \leq 0 \\ -\infty & else, \end{cases} \tag{37}$$

**Definition 2** ($\mathbf{L}_{D,h}$). *Define a functional $\mathbf{L}_{D,h}$ associated to* (30) *as*

$$\mathbf{L}_{D,h}^{\alpha, \beta}(\boldsymbol{f}) = \begin{cases} \int_\Omega \boldsymbol{f} d\alpha - \int_\Omega \boldsymbol{f} d\beta - h m_\beta & \boldsymbol{f} \in Lip(\Omega) , \ -h \leq \boldsymbol{f} \leq 0, \\ -\infty & else \end{cases}$$

With this definition, (30) becomes

$$\mathcal{L}_{D,h}(\alpha,\beta) = \sup_{\boldsymbol{f} \in C(\Omega)} \mathbf{L}_{D,h}^{\alpha,\beta}(\boldsymbol{f})$$

Problem (34) and (35) become

$$\mathcal{L}_{M,m}(\alpha,\beta) = \sup_{\boldsymbol{f} \in C(\Omega), h \in \mathbb{R}} \mathbf{L}_{M,m}^{\alpha,\beta}(\boldsymbol{f},h) \ \ and \ \ \mathcal{L}_{M,m}(\alpha,\beta) = \sup_{\boldsymbol{f} \in C(\Omega)} \overline{\mathbf{L}_{M,m}^{\alpha,\beta}}(\boldsymbol{f})$$

respectively.

We remark that the maximizer of $\mathbf{L}_{D,h}^{\alpha,\beta}$ exists.

**Proposition 3** (Existence of optimizer Schmitzer & Wirth (2019))**.** *For $\alpha,\beta \in \mathcal{M}_+(\Omega)$ and $h > 0$, there exists $\boldsymbol{f} \in C(\Omega)$ such that $\mathcal{L}_{D,h}(\alpha,\beta) = \mathbf{L}_{D,h}^{\alpha,\beta}(\boldsymbol{f})$.*

Finally, we summarize the formulations discussed in this section in Table 1. Note that we have two equivalent KR forms of $\mathcal{L}_{M,m}$. Although $\overline{\mathbf{L}_{M,m}}$ has a simpler form without the extra variable $h$, it contains the infimum which is hard to implemented in practice. Thus we mostly use $\mathbf{L}_{M,m}$ in practical implementation.

Table 1: Equivalent formulations of $\mathcal{L}_{M,m}$ and $\mathcal{L}_{D,h}$

| | $\mathcal{L}_{M,m}$ | $\mathcal{L}_{D,h}$ |
|---|---|---|
| Primal | $\inf_{\pi} \int_{\Omega \times \Omega} c(x,y) d\pi(x,y)$ 
 $s.t.\ \pi(A \times \Omega) \le \alpha(A),\ \forall A,$ 
 $\pi(\Omega \times A) \le \beta(A),\ \forall A,$ 
 $\pi(\Omega \times \Omega) \ge m,$ | $\inf_{\pi} \int_{\Omega \times \Omega} c(x,y) d\pi(x,y) - hm(\pi)$ 
 $s.t.\ \pi(A \times \Omega) \le \alpha(A),\ \forall A,$ 
 $\pi(\Omega \times A) \le \beta(A),\ \forall A,$ |
| Dual | $\sup_{(\boldsymbol{f},\boldsymbol{g},h)} \int_{\Omega} \boldsymbol{f} d\alpha + \int_{\Omega} \boldsymbol{g} d\beta + mh$ 
 $s.t.\ (\boldsymbol{f},\boldsymbol{g},h) \in C(\Omega) \times C(\Omega) \times \mathbb{R},$ 
 $\boldsymbol{f} \le 0,\ \boldsymbol{g} \le 0,\ h \ge 0$ 
 $c(x,y) - h - \boldsymbol{f}(x) - \boldsymbol{g}(y) \ge 0, \forall x,y$ | $\sup_{(\boldsymbol{f},\boldsymbol{g})} \int_{\Omega} \boldsymbol{f} d\alpha + \int_{\Omega} \boldsymbol{g} d\beta$ 
 $s.t.\ (\boldsymbol{f},\boldsymbol{g}) \in C(\Omega) \times C(\Omega),$ 
 $\boldsymbol{f} \le 0,\ \boldsymbol{g} \le 0,$ 
 $c(x,y) - h - \boldsymbol{f}(x) - \boldsymbol{g}(y) \ge 0, \forall x,y$ |
| KR $(c=d)$ | $\sup_{\boldsymbol{f},h} \int_{\Omega} \boldsymbol{f} d(\alpha - \beta) + h(m - m_\beta)$ 
 $s.t.\ \boldsymbol{f} \in Lip(\Omega), h \ge 0$ 
 $-h \le \boldsymbol{f} \le 0$ 

 $\sup_{\boldsymbol{f}} \int_{\Omega} \boldsymbol{f} d(\alpha - \beta) - \inf(\boldsymbol{f})(m - m_\beta)$ 
 $s.t.\ \boldsymbol{f} \in Lip(\Omega)$ 
 $\boldsymbol{f} \le 0$ | $\sup_{\boldsymbol{f}} \int_{\Omega} \boldsymbol{f} d(\alpha - \beta) - hm_\beta$ 
 $s.t.\ \boldsymbol{f} \in Lip(\Omega)$ 
 $-h \le \boldsymbol{f} \le 0$ |

## D  DIFFERENTIABILITY

This section proves the differentiability of both $\mathcal{L}_{D,h}$ and $\mathcal{L}_{M,m}$ in the KR form in Proposition 6 and Proposition 5. To this end, we first need to show the potential of $\mathcal{L}_{M,m}$ exists in Sec.D.1,

### D.1 EXISTENCE OF OPTIMIZERS

In this section, we prove that the maximizer of $\mathbf{L}_{M,m}^{\alpha,\beta}$ exists.

**Proposition 4** (Existence of optimizers). *For $\alpha, \beta \in \mathcal{M}_+(\Omega)$ and $m > 0$, there exists $\boldsymbol{f} \in C(\Omega)$ and $h \in \mathbb{R}$ such that $\mathcal{L}_{M,m}(\alpha,\beta) = \mathbf{L}_{M,m}^{\alpha,\beta}(\boldsymbol{f},h)$.*

To prove this proposition, we need the following lemma.

**Lemma 1** (Continuity). $\mathbf{L}_{M,m}^{\alpha,\beta}$ *is continuous on $C(\Omega) \times \mathbb{R}$.*

*Proof.* Let $(\boldsymbol{f}_n, h_n) \to (\boldsymbol{f}, h)$ in $C(\Omega) \times \mathbb{R}$. Assume $\mathbf{L}_{M,m}^{\alpha,\beta}(\boldsymbol{f}_n, h_n) > -\infty$ when $n$ is sufficiently large. We first check $\mathbf{L}_{M,m}^{\alpha,\beta}(\boldsymbol{f}, h) > -\infty$ as follows. For arbitrary $\epsilon > 0$, there exists $N > 0$ such that for $n > N$, $h_n < h + \epsilon$, thus $\boldsymbol{f}_n > -h_n > -h - \epsilon$. By taking $n \to \infty$, we see for arbitrary $\epsilon > 0$, $\boldsymbol{f} > -h - \epsilon$, which suggests $\boldsymbol{f} \geq -h$. In addition, it is easy to see $\boldsymbol{f} \leq 0$ and $h \geq 0$. It is also easy to see $Lip(\boldsymbol{f}) \leq 1$ due to the closeness of $Lip(\Omega)$. Thus according to the definition 36, we claim $\mathbf{L}_{M,m}^{\alpha,\beta}(\boldsymbol{f}, h) > -\infty$. Furthermore, since $-h - \epsilon < \boldsymbol{f}_n < 0$, and $\boldsymbol{f}_n \to \boldsymbol{f}$, by dominated convergence theorem, we have

$$\lim_{n\to\infty} \int_\Omega \boldsymbol{f}_n d\alpha = \int_\Omega \lim_{n\to\infty} \boldsymbol{f}_n d\alpha = \int_\Omega \boldsymbol{f} d\alpha, \tag{38}$$

$$and \lim_{n\to\infty} \int_\Omega \boldsymbol{f}_n d\beta = \int_\Omega \lim_{n\to\infty} \boldsymbol{f}_n d\beta = \int_\Omega \boldsymbol{f} d\beta. \tag{39}$$

Note we have $h_n(m - m_\beta) \to h(m - m_\beta)$. We conclude the proof by combining these three terms and obtaining $\mathbf{L}_{M,m}^{\alpha,\beta}(\boldsymbol{f}_n, h_n) \to \mathbf{L}_{M,m}^{\alpha,\beta}(\boldsymbol{f}, h)$. □

*Proof of Proposition 4.* If we can find a maximizing sequence $(\boldsymbol{f}_n, h_n)$ that converges to $(\boldsymbol{f}, h) \in C(\Omega) \times \mathbb{R}$, then Lemma 1 suggests that $\mathcal{L}_{M,m}(\alpha,\beta) = \sup_{\boldsymbol{f},h} \mathbf{L}_{M,m}^{\alpha,\beta}(\boldsymbol{f},h) = \lim_{n\to\infty} \mathbf{L}_{M,m}^{\alpha,\beta}(\boldsymbol{f}_n, h_n) = \mathbf{L}_{M,m}^{\alpha,\beta}(\boldsymbol{f}, h)$, which proves this proposition. Therefore, we only need to show that it is always possible to construct such a maximizing sequence.

Let $(\boldsymbol{f}_n, h_n)$ be a maximizing sequence. We abbreviate $\max(\boldsymbol{f}_n) = \max_{x\in\Omega}(\boldsymbol{f}_n(x))$ and $\min(\boldsymbol{f}_n) = \min_{x\in\Omega}(\boldsymbol{f}_n(x))$. We first assume $\boldsymbol{f}_n$ does not have any bounded subsequence, then there exists $N > 0$, such that for all $n > N$, $\min(\boldsymbol{f}_n) < -diam(\Omega)$ (otherwise we can simply collect a subsequence of $\boldsymbol{f}_n$ bounded by $diam(\Omega)$). We can therefore construct $\widetilde{\boldsymbol{f}_n} = \boldsymbol{f}_n - (\min(\boldsymbol{f}_n) + diam(\Omega))$ and $\widetilde{h_n} = h_n + (\min(\boldsymbol{f}_n) + diam(\Omega))$. Note that $\max(\widetilde{\boldsymbol{f}_n}) \leq \min(\widetilde{\boldsymbol{f}_n}) + diam(\Omega) = -diam(\Omega) + diam(\Omega) = 0$, $\widetilde{\boldsymbol{f}_n} + \widetilde{h_n} = \boldsymbol{f}_n + h_n \geq 0$, and $\widetilde{\boldsymbol{f}_n} \in Lip(\Omega)$, so $\mathbf{L}_{M,m}^{\alpha,\beta}(\widetilde{\boldsymbol{f}_n}, \widetilde{h_n}) > -\infty$, and

$$\mathbf{L}_{M,m}^{\alpha,\beta}(\widetilde{\boldsymbol{f}_n}, \widetilde{h_n}) = \int_\Omega \widetilde{\boldsymbol{f}_n} d(\alpha - \beta) + \widetilde{h_n}(m - m_\beta)$$

$$= \int_\Omega \boldsymbol{f}_n d(\alpha - \beta) + h_n(m - m_\beta) + (\min(\boldsymbol{f}_n) - diam(\Omega))(m - m_\alpha)$$

$$\geq \mathbf{L}_{M,m}^{\alpha,\beta}(\boldsymbol{f}_n, h_n),$$

which suggests that $(\widetilde{\boldsymbol{f}_n}, \widetilde{h_n})$ is a better maximizing sequence than $(\boldsymbol{f}_n, h_n)$. Note $\widetilde{\boldsymbol{f}_n}$ is uniformly bounded by $diam(\Omega)$ because $0 \geq \widetilde{\boldsymbol{f}_n} \geq \min(\widetilde{\boldsymbol{f}_n}) = -diam(\Omega)$. As a result, we can always assume $\widetilde{h_n}$ is also bounded by $diam(\Omega)$. Because otherwise we can construct $\overline{h_n} = -\min(\widetilde{\boldsymbol{f}_n}) \leq diam(\Omega)$, and it is easy to show $(\widetilde{\boldsymbol{f}_n}, \overline{h_n})$ is a better maximizing sequence than $(\widetilde{\boldsymbol{f}_n}, \widetilde{h_n})$. In summary, we can always find a maximizing sequence $(\boldsymbol{f}_n, h_n)$, such that both $\boldsymbol{f}_n$ and $h_n$ are bounded by $diam(\Omega)$.

Finally, since $\boldsymbol{f}_n$ is uniformly bounded and equicontinuous, $\boldsymbol{f}_n$ converges uniformly (up to a subsequence) to a continuous function $\boldsymbol{f}$. In addition, $h_n$ has a convergent subsequence since it is bounded. Therefore, we can always find a maximizing sequence $(\boldsymbol{f}_n, h_n)$ that converges to some $(\boldsymbol{f}, h) \in C(\Omega) \times \mathbb{R}$, which finishes the proof. □

**Remark** The proof of Proposition 4 is an analogue of Proposition 2.11 in Schmitzer & Wirth (2019). The difference is that in our Proposition 4, besides $\boldsymbol{f}$, we need to handle another variable $h$ acting as the lower bound of $\boldsymbol{f}$. In contrast, Proposition 2.11 in Schmitzer & Wirth (2019) only handles the fixed $h$.

### D.2 COMPUTATION OF GRADIENT

As we have proved the existence of the potential of $\mathcal{L}_{M,m}$, we can now consider the differentiability of $\mathcal{L}_{M,m}$ in the KR form.

We consider a transformation $\mathcal{T} : \mathbb{R}^d \times \Omega \to \Omega$ parametrized by $\theta \in \mathbb{R}^d$. Let $\mathcal{T}_\theta(\cdot) = \mathcal{T}(\theta, \cdot)$ and denote $\beta_\theta = \mathcal{T}_\theta(\beta)$ the corresponding push-forward measure of $\beta$. We show in Proposition 6 and 5 that, if $\mathcal{T}_\theta$ is Lipschitz w.r.t. $\theta$, then the objective functions $\mathcal{L}_{D,h}(\alpha, \beta_\theta)$ and $\mathcal{L}_{M,m}(\alpha, \beta_\theta)$ is differentiable w.r.t. $\theta$, and the gradient can be computed explicitly.

**Proposition 5** (Differentiability of $\mathcal{L}_{M,m}$)**.** *If $\mathcal{T}_\theta : \Omega \to \Omega$ is Lipschitz w.r.t. to $\theta$, then $\mathcal{L}_{M,m}(\alpha, \beta_\theta)$ is continuous w.r.t. $\theta$, and is differentiable almost everywhere. Furthermore, we have*

$$\nabla_\theta \mathcal{L}_{M,m}(\alpha, \beta_\theta) = -\int_\Omega \nabla_\theta \boldsymbol{f}(\mathcal{T}_\theta(x))d\beta, \tag{40}$$

*where $(\boldsymbol{f}, h)$ is a maximizer of $\mathbf{L}_{M,m}$.*

*Proof.* To begin with, for arbitrary $\theta, \theta'$, we consider the following two-step procedure that transports $m$ unit mass from $\alpha$ to $\beta_{\theta'}$. First, we transport $m$ mass from $\alpha$ to $\beta_\theta$ according to $\overline{\pi}$, which is a solution to $\mathcal{L}_{M,m}(\alpha, \beta_\theta)$. Second, we transport $m$ mass received in $\beta_\theta$ to $\beta_{\theta'}$ according to a plan $\pi'$ defined as

$$\pi'(A \times B) = \begin{cases} \widetilde{\pi}(A \times B)\frac{\overline{\pi}_\#^2(A)}{\beta_\theta(A)} & if \ \beta_\theta(A) > 0 \\ 0 & else, \end{cases}$$

where $\widetilde{\pi}(\mathcal{T}_\theta(x), \mathcal{T}_{\theta'}(x)) = \beta(x)$. That is to transport all mass received at $\mathcal{T}_\theta(x)$ to the corresponding point $\mathcal{T}_{\theta'}(x)$. Thus we have

$$\int_{\Omega \times \Omega} d(x,z)d\pi'(x,z) \leq \int_{\Omega \times \Omega} d(x,z)d\widetilde{\pi}(x,z) = \int_\Omega d(\mathcal{T}_\theta(x), \mathcal{T}_{\theta'}(x))d\beta(x).$$

Since $\mathcal{T}_\theta$ is Lipschitz w.r.t. $\theta$, there exists a constant $\Delta > 0$, such that $d(\mathcal{T}_\theta(x), \mathcal{T}_{\theta'}(x)) \leq \Delta||\theta - \theta'||$. Therefore, the cost of the second step can be bounded as

$$\int_{\Omega \times \Omega} d(x,z)d\pi'(x,z) \leq m_\beta \Delta||\theta - \theta'||.$$

Denote $cost_1$ the overall cost of this two-step procedure. We have

$$cost_1 \leq \mathcal{L}_{M,m}(\alpha, \beta_\theta) + m_\beta \Delta||\theta - \theta'||.$$

By applying the gluing lemma (Villani (2009) Sec.1) to $\overline{\pi}$ and $\pi'$, we can construct a transport plan $\widetilde{\pi}$ that transports $m$ unit mass from $\overline{\pi}_\#^1$ to $\pi_\#^{'2}$. Denote $cost_2 = \int_{\Omega \times \Omega} d(x,y)d\widetilde{\pi}(x,y)$ the cost of $\widetilde{\pi}$. On one hand, we have

$$\mathcal{L}_{M,m}(\alpha, \beta_{\theta'}) \leq cost_2$$

according to the definition of $\mathcal{L}_{M,m}$. On the other hand, we have

$$cost_2 \leq cost_1.$$

Because for arbitrary $x, z \in \Omega$, the two-step procedure and $\widetilde{\pi}$ transport the same amount of mass between them. However, the cost of transporting a unit mass from $x$ to $z$ is $d(x,y) + d(y,z)$ for a $y \in \Omega$ for the two-step procedure, but is $d(x,z)$ for $\widetilde{\pi}$, which is cheaper according to triangle inequality. By combining these inequalities together, we have

$$\mathcal{L}_{M,m}(\alpha, \beta_{\theta'}) \leq cost_2 \leq cost_1 \leq \mathcal{L}_{M,m}(\alpha, \beta_\theta) + m_\beta \Delta||\theta - \theta'||,$$

*i.e.,*

$$\mathcal{L}_{M,m}(\alpha, \beta_{\theta'}) \leq \mathcal{L}_{M,m}(\alpha, \beta_\theta) + m_\beta \Delta||\theta - \theta'||.$$

By switching $\theta$ and $\theta'$ and repeating the argument, we have

$$\mathcal{L}_{M,m}(\alpha, \beta_\theta) \le \mathcal{L}_{M,m}(\alpha, \beta_{\theta'}) + m_\beta \Delta ||\theta - \theta'||,$$

thus we have

$$|\mathcal{L}_{M,m}(\alpha, \beta_\theta) - \mathcal{L}_{M,m}(\alpha, \beta_{\theta'})| \le m_\beta \Delta ||\theta - \theta'||,$$

which suggests $\mathcal{L}_{M,m}(\alpha, \beta_\theta)$ is continuous w.r.t. $\theta$, and Radamacher's theorem states that it is differentiable almost everywhere.

The sketch of the rest of the proof is as follows. According to Proposition 4, the maximizer to the KR form of $\mathcal{L}_{M,m}(\alpha, \beta_\theta)$ exists, thus we can write $\nabla_\theta \mathcal{L}_{M,m}(\alpha, \beta_\theta)$ as $-\nabla_\theta \int_\Omega \boldsymbol{f}(\mathcal{T}_\theta(x)) d\beta$ according to the envelope theorem (Milgrom & Segal, 2002). Then we prove

$$\nabla_\theta \int_\Omega \boldsymbol{f}(\mathcal{T}_\theta(x)) d\beta = \int_\Omega \nabla_\theta \boldsymbol{f}(\mathcal{T}_\theta(x)) d\beta,$$

when the right hand side of the equation is well defined following Arjovsky et al. (2017), which completes our proof. $\square$

**Proposition 6** (Differentiability of $\mathcal{L}_{D,h}$). *If $\mathcal{T}_\theta : \Omega \to \Omega$ is Lipschitz w.r.t. to $\theta$, then $\mathcal{L}_{D,h}(\alpha, \beta_\theta)$ is continuous w.r.t. $\theta$, and is differentiable almost everywhere. Furthermore, we have*

$$\nabla_\theta \mathcal{L}_{D,h}(\alpha, \beta_\theta) = -\int_\Omega \nabla_\theta \boldsymbol{f}(\mathcal{T}_\theta(x)) d\beta, \tag{41}$$

*where $\boldsymbol{f}$ is a maximizer of $\mathbf{L}_{D,h}$.*

*Proof.* We first prove $\mathcal{L}_{D,h}(\alpha, \beta_\theta)$ is continuous w.r.t. $\theta$. By definition, we have $\mathcal{L}_{D,h}(\alpha, \beta) = KR_h(\alpha, \beta) - \frac{h}{2}(m_\alpha + m_\beta)$. Thus by triangle inequality of the KR metric, for arbitrary $\theta, \theta' \in \mathbb{R}^d$, we have

$$
\begin{aligned}
&|\mathcal{L}_{D,h}(\alpha, \beta_\theta) - \mathcal{L}_{D,h}(\alpha, \beta_{\theta'})| \\
=&|\Big(KR_h(\alpha, \beta_\theta) - \frac{h}{2}(m_\alpha + m_{\beta_\theta})\Big) - \Big(KR_h(\alpha, \beta_{\theta'}) - \frac{h}{2}(m_\alpha + m_{\beta_{\theta'}})\Big)| \\
=&|KR_h(\alpha, \beta_\theta) - KR_h(\alpha, \beta_{\theta'})| \\
\le& KR_h(\beta_{\theta'}, \beta_\theta)
\end{aligned}
\tag{42}
$$

Note that the second equality holds because $\mathcal{T}$ maintains the total mass, *i.e.*, $m_{\beta_{\theta'}} = m_{\beta_\theta} = m_\beta$. In addition, we have

$$KR_h(\beta_{\theta'}, \beta_\theta) \le \mathcal{W}_1(\beta_{\theta'}, \beta_\theta) \le \int_\Omega d(\mathcal{T}_\theta(y) - \mathcal{T}_{\theta'}(y)) d\beta(y).$$

By assumption, $\mathcal{T}_\theta$ is Lipschitz w.r.t. $\theta$, thus there exists a constant $\Delta > 0$, such that $d(\mathcal{T}_\theta(y), \mathcal{T}_{\theta'}(y)) \le \Delta ||\theta - \theta'||$. Thus $KR_h(\beta_{\theta'}, \beta_\theta) \le m_\beta \Delta ||\theta - \theta'||$. Finally, we have

$$\mathcal{L}_{D,h}(\alpha, \beta_\theta) - \mathcal{L}_{D,h}(\alpha, \beta_{\theta'}) \le KR_h(\beta_{\theta'}, \beta_\theta) \le m_\beta \Delta ||\theta - \theta'||,$$

which proves $\mathcal{L}_{D,h}(\alpha, \beta_\theta)$ is continuous w.r.t. $\theta$, and Radamacher's theorem states that it is differentiable almost everywhere.

The rest of the proof is similar to that of Proposition 5. $\square$

**Remark** The main idea used in the proofs in this section is based on that in Arjovsky et al. (2017). However, a key difference is that unlike $\mathcal{W}_1$, neither $\mathcal{L}_{M,m}$ nor $\mathcal{L}_{D,h}$ is a metric, *i.e.*, they do not necessarily satisfy triangle inequality, thus the differences cannot be bounded directly. This gap is bridged by using the KR metric in Proposition 6, and by constructing a transportation plan in Proposition 5.

### D.3 Application to Point Registration

For point set registration task, we focus on the discrete distributions and re-state the previous results. We first present the proof of the main theorems in the main text.

*Proof of Theorem 1.* The KR formulation of $\mathcal{L}_{D,h}$ is given in (30), and the existence of the optimizer is proved in Proposition 3. The gradient of $\mathcal{L}_{D,h}(\alpha, \beta_\theta)$ is derived in Proposition 6. Theorem 1 is proved by applying these results to discrete distributions. □

*Proof of Theorem 2.* The KR formulation of $\mathcal{L}_{M,m}$ is given in Theorem 3, and the existence of the optimizer is proved in Proposition 4. The gradient of $\mathcal{L}_{M,m}(\alpha, \beta_\theta)$ is derived in Proposition 5. Theorem 2 is proved by applying these results to discrete distributions. □

Then we verify that the parametrized transformation $\mathcal{T}_\theta$ used in our paper satisfies the regularity assumption, *i.e.*, it is Lipschitz w.r.t. the parameter $\theta$.

**Lemma 2.** *If $\mathcal{T} : \mathbb{R}^d \times \Omega \to \Omega$ is Lipschitz w.r.t. to its first variable, then for arbitrary $y \in Y$, and $\theta$, $\theta' \in \mathbb{R}^d$, there exists $\Delta > 0$, such that*

$$||\mathcal{T}_\theta(y) - \mathcal{T}_{\theta'}(y)|| \le \Delta ||\theta - \theta'||.$$

**Proposition 7.** *Given a point set $Y = \{y_i\}_{i=1}^r \subseteq \Omega$. Let $\theta = (A, t, V) \in \mathbb{R}^{12+3r}$, where $A \in \mathbb{R}^{3\times3}$ is a affinity matrix, $t \in \mathbb{R}^{1\times3}$ is a translation vector, $V \in \mathbb{R}^{r\times3}$ is the offset vectors for all points in $Y$, and $V_i$ represent the $i$-th row of $V$. For each point $y_i \in Y$, define $\mathcal{T}_\theta(y_i) = y_i A + t + V_i$, where $V_i$ represents the $i$-th row of $V$. $\mathcal{T} : \mathbb{R}^d \times \Omega \to \Omega$ is Lipschitz w.r.t. to its first variable.*

*Proof.* Note for arbitrary $A$, there exists $\delta > 0$, such that $||A||_2 \le \delta ||A||_F$, where $|| \cdot ||_F$ is the Frobenius norm. For arbitrary $\theta = (A, t, V), \widetilde{\theta} = (\widetilde{A}, \widetilde{t}, \widetilde{V})$ and $y_i \in Y$, we have

$$
\begin{aligned}
||\mathcal{T}_\theta(y_i) - \mathcal{T}_{\widetilde{\theta}}(y_i)||_2 &= ||(A - \widetilde{A})y_i + (t - \widetilde{t}) + (V_i - \widetilde{V}_i)||_2 \\
&\le ||A - \widetilde{A}||_2 ||y_i||_2 + ||t - \widetilde{t}||_2 + ||V_i - \widetilde{V}_i||_2 \\
&\le \delta ||A - \widetilde{A}||_F ||y_i||_2 + ||\theta - \widetilde{\theta}||_2 + ||\theta - \widetilde{\theta}||_2 \\
&\le \delta ||\theta - \widetilde{\theta}||_2 ||y_i||_2 + ||\theta - \widetilde{\theta}||_2 + ||\theta - \widetilde{\theta}||_2 \\
&= (\delta ||y_i|| + 2) ||\theta - \widetilde{\theta}||_2,
\end{aligned}
$$

which proves that $\mathcal{T}$ is Lipschitz w.r.t. to its first variable. □

## E Properties

This section answers two questions: 1) What are the connections between KR forms of $\mathcal{L}_{M,m}$ and $\mathcal{L}_{D,h}$? 2) What does the potential of $\mathcal{L}_{M,m}$ looks like? We briefly discuss the first question in Sec. E.1. For the second question, the main result is Proposition 11 in Sec. E.2, where we show that for each point in $\alpha$ and $\beta$, the potential $\boldsymbol{f}$ either has gradient norm 1 or attains its maximum or minimum.

### E.1 Connections between $\mathcal{L}_M$, $\mathcal{L}_D$ and $\mathcal{W}_1$

This subsection briefly discuss the relations between the KR forms of $\mathcal{L}_{M,m}$, $\mathcal{L}_{D,h}$ and $\mathcal{W}_1$.

The following proposition presents a simple fact about the relation between $\mathcal{L}_M$ and $\mathcal{L}_D$. We omit all proves in this subsection as all results can be easily verified.

**Proposition 8** (Relations between $\mathcal{L}_M$ and $\mathcal{L}_D$). *Let $\boldsymbol{f}$ be a maximizer of $\overline{\mathbf{L}_{M,m}^{\alpha,\beta}}$. For the fixed $h^* = -\inf(\boldsymbol{f})$, $\boldsymbol{f}$ is also a maximizer of $\mathbf{L}_{D,h^*}^{\alpha,\beta}$.*

Now we turn to the special cases of $\mathcal{L}_{M,m}$ and $\mathcal{L}_{D,h}$. To begin with, we define two useful functionals as follows.

**Definition 3** ($\mathbf{L}_P^{\alpha,\beta}$). *Define functional $\mathbf{L}_P^{\alpha,\beta}$ as*

$$\mathbf{L}_P^{\alpha,\beta}(\boldsymbol{f}) = \begin{cases} \int_\Omega \boldsymbol{f} d\alpha - \int_\Omega \boldsymbol{f} d\beta & \boldsymbol{f} \in Lip(\Omega), \ \boldsymbol{f} \leq 0, \ m_\alpha \geq m_\beta \\ -\infty & else, \end{cases} \tag{43}$$

*and define $\mathcal{L}_P(\alpha,\beta) = \sup_{\boldsymbol{f} \in C(\Omega)} \mathbf{L}_P^{\alpha,\beta}(\boldsymbol{f})$.*

**Definition 4** ($\mathbf{W}_1^{\alpha,\beta}$). *Define the functional associated to Wasserstein-1 metric $\mathcal{W}_1$ as*

$$\mathbf{L}_W^{\alpha,\beta}(\boldsymbol{f}) = \begin{cases} \int_\Omega \boldsymbol{f} d\alpha - \int_\Omega \boldsymbol{f} d\beta & \boldsymbol{f} \in Lip(\Omega), \ m_\alpha \geq m_\beta \\ -\infty & else, \end{cases} \tag{44}$$

*thus $\mathcal{W}_1$ can be expressed as $\mathcal{W}_1(\alpha,\beta) = \sup_{\boldsymbol{f} \in C(\Omega)} \mathbf{L}_W^{\alpha,\beta}(\boldsymbol{f})$.*

The following proposition describes the relations between $\mathcal{L}_{M,m}$, $\mathcal{L}_{D,h}$, $\mathcal{L}_P$ and $\mathcal{W}_1$.

**Proposition 9** (Special cases of $\mathcal{L}_{M,m}$ and $\mathcal{L}_{D,h}$). *(1) When $m_\alpha \geq m_\beta = m$, the maximizer of $\mathbf{L}_{M,m}$ is also a maximizer of $\mathbf{L}_P$.*

*(2) When $m_\alpha \geq m_\beta$ and $h \geq diam(\Omega)$, the maximizer of $\mathbf{L}_{D,h}$ is also a maximizer of $\mathbf{L}_P$.*

*(3) When $m_\alpha = m_\beta$, the maximizer of $\mathbf{L}_P$ is also a maximizer of $\mathbf{W}_1$.*

We note $\mathcal{L}_P$ is for "semi-complete" Wasserstein problem, where all mass of $\beta$ is transported to $\alpha$. Thus it may be interesting on its own, for example in the template matching problem where the data points in an incomplete "data distribution" is matched to a complete "model distribution".

Finally, we have the following straightforward corollary.

**Corollary 1.** *(1) When $m_\alpha = m_\beta = m$, $\mathcal{L}_{M,m}$ is equivalent to $\mathcal{W}_1$.*

*(2) When $m_\alpha = m_\beta$ and $h \geq diam(\Omega)$, $\mathcal{L}_{D,h}$ is equivalent to $\mathcal{W}_1 - hm_\beta$.*

### E.2 PROPERTIES OF THE POTENTIALS

Consider $\mathbf{W}_1$ as a special case of $\overline{\mathbf{L}_{M,m}}$. Its potential has the following property.

**Lemma 3** (Potential of $\mathcal{W}_1$ (Gulrajani et al., 2017)). *Let $\boldsymbol{f}$ be a maximizer of $\mathbf{W}_1^{\alpha,\beta}$. Then $\boldsymbol{f}$ has gradient norm 1 $(\alpha + \beta)$-almost surely.*

The main result of this subsection is the extension of this property to $\overline{\mathbf{L}_{M,m}}$ in Proposition 11.

To begin with, we note that by definition, $\mathcal{L}_{M,m}(\alpha,\beta)$ only transports a a fraction of mass and discards the other. We called the transported mass *"active"* and the discarded mass *"inactive"*. An important observation is that, if we throw away some inactive mass, the solution to the problem will not be affected; and if we only focus on the active mass, we immediately obtain a solution to $\mathcal{W}_1$. This is formally stated as follows.

**Proposition 10.** *Let $\pi$ be the solution to the primal form of $\mathcal{L}_{M,m}(\alpha,\beta)$. Define $\alpha'$ and $\beta'$ be measures satisfying*

$$\pi_\#^1(A) \leq \alpha'(A) \leq \alpha(A) \quad and \quad \pi_\#^2(A) \leq \beta'(A) \leq \beta(A). \tag{45}$$

*for an arbitrary measurable set $A$.*

*(1) $\pi$ is also the solution to the primal form of $\mathcal{L}_{M,m}(\alpha',\beta')$ and $\mathcal{W}_1(\pi_\#^1, \pi_\#^2)$, thus*

$$\mathcal{L}_{M,m}(\alpha,\beta) = \mathcal{L}_{M,m}(\alpha',\beta') = \mathcal{W}_1(\pi_\#^1, \pi_\#^2) \tag{46}$$

*(2) Let $\boldsymbol{f}$ be a maximizer of $\overline{\mathbf{L}_{M,m}^{\alpha,\beta}}$. Then $\boldsymbol{f}$ is also a maximizer of $\overline{\mathbf{L}_{M,m}^{\alpha',\beta'}}$ and $\mathbf{W}_1^{\pi_\#^1, \pi_\#^2}$.*

*Proof.* (1) Notice all admissible solutions to $\mathcal{W}_1(\pi_\#^1, \pi_\#^2)$ are also admissible solutions to $\mathcal{L}_{M,m}(\alpha,\beta)$. Then we can prove $\mathcal{W}_1(\pi_\#^1, \pi_\#^2) = \mathcal{L}_{M,m}(\alpha,\beta)$ by contradiction. Similarly, we can prove $\mathcal{W}_1(\pi_\#^1, \pi_\#^2) = \mathcal{L}_{M,m}(\alpha,\beta)$.

(2) Note we have

$$
\begin{aligned}
\overline{\mathbf{L}_{M,m}^{\alpha',\beta'}}(\boldsymbol{f}) &= \int_\Omega \boldsymbol{f} d\alpha' + \int_\Omega (\inf(\boldsymbol{f}) - \boldsymbol{f}) d\beta' - \inf(\boldsymbol{f})m \\
&\geq \int_\Omega \boldsymbol{f} d\alpha + \int_\Omega (\inf(\boldsymbol{f}) - \boldsymbol{f}) d\beta - \inf(\boldsymbol{f})m = \overline{\mathbf{L}_{M,m}^{\alpha,\beta}}(\boldsymbol{f}) = \mathcal{L}_{M,m}(\alpha,\beta), \quad (47)
\end{aligned}
$$

where the inequality holds because $\boldsymbol{f} \leq 0$, $\inf(\boldsymbol{f}) - \boldsymbol{f} \leq 0$, $\alpha' \leq \alpha$ and $\beta' \leq \beta$. According to the first part of this proof, we have $\mathcal{L}_{M,m}(\alpha',\beta') = \mathcal{L}_{M,m}(\alpha,\beta)$, thus $\overline{\mathbf{L}_{M,m}^{\alpha',\beta'}}(\boldsymbol{f}) \leq \mathcal{L}_{M,m}(\alpha',\beta') = \mathcal{L}_{M,m}(\alpha,\beta)$. By combining these two equalities, we conclude that $\overline{\mathbf{L}_{M,m}^{\alpha',\beta'}}(\boldsymbol{f}) = \mathcal{L}_{M,m}(\alpha',\beta') = \mathcal{L}_{M,m}(\alpha,\beta)$, *i.e.*, $\boldsymbol{f}$ is a maximizer of $\overline{\mathbf{L}_{M,m}^{\alpha',\beta'}}$.

In addition, we have

$$
\begin{aligned}
\mathbf{W}_1^{\pi_\#^1,\pi_\#^2}(\boldsymbol{f}) &= \int_\Omega \boldsymbol{f} d\pi_\#^1 - \int_\Omega \boldsymbol{f} d\pi_\#^2 \geq \int_\Omega \boldsymbol{f} d\pi_\#^1 - \int_\Omega \boldsymbol{f} d\pi_\#^2 - \inf(\boldsymbol{f})(m - m(\pi_\#^1)) \\
&= \overline{\mathbf{L}_{M,m}^{\pi_\#^1,\pi_\#^2}}(\boldsymbol{f}) \geq \overline{\mathbf{L}_{M,m}^{\alpha,\beta}}(\boldsymbol{f}) = \mathcal{L}_{M,m}(\alpha,\beta),
\end{aligned}
$$

where the first inequality holds because $-\inf(\boldsymbol{f})(m - m(\pi_\#^1)) \leq 0$, and the second inequality holds following equation (47). According to the first part of this proof, we have $\mathcal{W}_1(\pi_\#^1, \pi_\#^2) = \mathcal{L}_{M,m}(\alpha,\beta)$, thus $\mathbf{W}_1^{\pi_\#^1,\pi_\#^2}(\boldsymbol{f}) \leq \mathcal{W}_1(\pi_\#^1, \pi_\#^2) = \mathcal{L}_{M,m}(\alpha,\beta)$. By combining these two equalities, we conclude $\mathbf{W}_1^{\pi_\#^1,\pi_\#^2}(\boldsymbol{f}) = \mathcal{W}_1(\pi_\#^1, \pi_\#^2) = \mathcal{L}_{M,m}(\alpha,\beta)$, *i.e.*, $\boldsymbol{f}$ is a maximizer of $\mathbf{W}_1^{\pi_\#^1,\pi_\#^2}$. $\quad\square$

Thanks to Proposition 10 and Lemma 3, we can now characterize the potential of $\mathcal{L}_{M,m}$ on active mass. This is formally stated in the following corollary.

**Corollary 2.** *Let $\pi$ be the solution to the primal form of $\mathcal{L}_{M,m}(\alpha,\beta)$, and $\boldsymbol{f}$ be a maximizer of $\overline{\mathbf{L}_{M,m}^{\alpha,\beta}}$. Then $\boldsymbol{f}$ has gradient norm $1$ $(\pi_\#^1 + \pi_\#^2)$-almost surely.*

Now, in order to completely characterize the potential of $\mathcal{L}_{M,m}$, we only need to characterize the potential on the inactive mass. In fact, we find that the behavior of potential is rather simple on the inactive mass, *i.e.*, it always attains its maximum or minimum. This is formally stated as follows.

**Corollary 3** (Flatness on inactive mass). *Let $\boldsymbol{f}$ be a maximizer of $\overline{\mathbf{L}_{M,m}^{\alpha,\beta}}$, and $\pi$ be the solution to the primal form of $\mathcal{L}_{M,m}^{\alpha,\beta}$. Let $\alpha_{\mathcal{S}}$ and $\beta_{\mathcal{S}}$ be non-negative measures satisfying*

$$
\pi_\#^1 \leq \alpha - \alpha_{\mathcal{S}} \leq \alpha \quad and \quad \pi_\#^2 \leq \beta - \beta_{\mathcal{S}} \leq \beta.
$$

*Then $\boldsymbol{f} = 0$ $\alpha_{\mathcal{S}}$-almost surely, and $\boldsymbol{f} = -\inf(\boldsymbol{f})$ $\beta_{\mathcal{S}}$-almost surely.*

*Proof.* According to Proposition 10, we have $\mathcal{L}_{M,m}^{\alpha-\alpha_{\mathcal{S}},\beta} = \mathcal{L}_{M,m}^{\alpha,\beta}$, and $\boldsymbol{f}$ is a maximizer of $\overline{\mathbf{L}_{M,m}^{\alpha-\alpha_{\mathcal{S}},\beta}}$. In other words, we have

$$
\int_\Omega \boldsymbol{f} d\alpha - \int_\Omega \boldsymbol{f}\beta - \inf(\boldsymbol{f})(m - m_\beta) = \int_\Omega \boldsymbol{f} d(\alpha - \alpha_{\mathcal{S}}) - \int_\Omega \boldsymbol{f}\beta - \inf(\boldsymbol{f})(m - m_\beta).
$$

By cleaning this equation, we obtain $\int_\Omega \boldsymbol{f} d\alpha_{\mathcal{S}} = 0$. Since $\boldsymbol{f} \leq 0$ and $\alpha_{\mathcal{S}}$ is a non-negative measure, we conclude that $\boldsymbol{f} = 0$ $\alpha_{\mathcal{S}}$-almost surely. The statement for $\beta_{\mathcal{S}}$ can be proved in a similar way. $\quad\square$

Finally, we can present a qualitative description for the potential of $\mathcal{L}_{M,m}$ by decompose mass $\alpha$ and $\beta$ into active and inactive mass.

**Proposition 11.** *Let $\boldsymbol{f}$ be a maximizer of $\mathbf{L}_{M,m}(\alpha,\beta)$. There exist non-negative measures $\mu$, $\nu_\alpha \leq \alpha$ and $\nu_\beta \leq \beta$ satisfying $\mu + \nu_\alpha + \nu_\beta = \alpha + \beta$, such that 1) $\boldsymbol{f}$ has gradient norm $1$ $\mu$-almost surely 2) $\boldsymbol{f}$ attains the maximum $\nu_\alpha$-almost surely, and 3) $\boldsymbol{f}$ attains the minimum $\nu_\beta$-almost surely.*

We note that similar conclusion holds for $\mathcal{L}_{D,h}$.

**Proposition 12.** *Let $\boldsymbol{f}$ be a maximizer of $\mathbf{L}_{D,h}(\alpha,\beta)$. There exist non-negative measures $\mu$, $\nu_\alpha \leq \alpha$ and $\nu_\beta \leq \beta$ satisfying $\mu + \nu_\alpha + \nu_\beta = \alpha + \beta$, such that 1) $\boldsymbol{f}$ has gradient norm $1$ $\mu$-almost surely 2) $\boldsymbol{f}$ attains the maximum $\nu_\alpha$-almost surely, and 3) $\boldsymbol{f}$ attains the minimum $\nu_\beta$-almost surely.*

**Remark**    Given Proposition 11, an important question is where are the inactive mass. An direct observation is that if a region is far away from one of the measures, then all mass within this region is inactive. Therefore, we can easily identify some inactive regions where the potential attains its maximum or minimum, *i.e.*, flat. We present the following straightforward propositions without proof. A practical example is shown in Fig. 9.

**Corollary 4** (Flat region). *Let $\boldsymbol{f}$ denote a maximizer of $\overline{\mathbf{L}_{M,m}^{\alpha,\beta}}$. Denote regions*

$$\mathcal{F}_\alpha = \Big\{ x | dist(x, supp(\beta)) > -\inf(\boldsymbol{f}) \Big\} \text{ and } \mathcal{F}_\beta = \Big\{ y | dist(y, supp(\alpha)) > -\inf(\boldsymbol{f}) \Big\}.$$

*Then $\boldsymbol{f} = 0$ $\alpha|_{\mathcal{F}_\alpha}$-almost surely, and $\boldsymbol{f} = \inf(\boldsymbol{f})$ $\beta|_{\mathcal{F}_\beta}$-almost surely.*

**Corollary 5** (Flat region). *Let $\boldsymbol{f}$ denote a maximizer of $\mathbf{L}_{D,h}^{\alpha,\beta}$. Denote regions*

$$\mathcal{F}_\alpha = \Big\{ x | dist(x, supp(\beta)) > h \Big\} \text{ and } \mathcal{F}_\beta = \Big\{ y | dist(y, supp(\alpha)) > h \Big\}.$$

*Then $\boldsymbol{f} = 0$ $\alpha|_{\mathcal{F}_\alpha}$-almost surely, and $\boldsymbol{f} = -h$ $\beta|_{\mathcal{F}_\beta}$-almost surely.*

# F    MORE DETAILS OF THE MAIN TEXT

## F.1    MORE DETAILS OF SEC. 2

Our method is related to Wasserstein generative adversarial network (WGAN) (Arjovsky et al., 2017), which is a popular method for large scale DM problems. A recent survey of the applications of WGAN can be found in Gui et al. (2020). WGAN efficiently optimizes the Wasserstein-1 metric by approximating the KR potential using a neural network. This technique is also used in our method. In this sense, our method directly generalizes WGAN to PDM problems.

## F.2    MORE DETAILS OF SEC. 4.2

We present a simple example comparing the primal and the KR forms in Fig. 9. We show the correspondence $\mathbf{P}$ obtained by the primal form in the 1-st row, where the black lines link the corresponding points. To estimate the KR forms, we parametrize the potential function $\boldsymbol{f}_{w,h}$ by a neural network shown in Fig. 10, and learn $\boldsymbol{f}_{w,h}$ by maximizing (10) or (11). The 2-nd row visualizes the learned potential function $\boldsymbol{f}_{w,h}$. The corresponding gradient norms $|\nabla \boldsymbol{f}_{w,h}|$ are shown in the 3-rd row. $|\nabla \boldsymbol{f}_{w,h}|$ are further visualized in the 4-th row, where the size of each point is approximately proportional to the gradient norm.

We further evaluate the precision of the estimated KR forms. For each setting, we compute $\mathcal{L}_{M,m}$ or $\mathcal{L}_{D,h}$ in the KR forms by inserting the learned potential $\boldsymbol{f}_{w,h}$ back into (10) or (11) (without the gradient penalty term). To obtain the true values of $\mathcal{L}_{M,m}$ and $\mathcal{L}_{D,h}$, we compute their primal forms (1) and (2) using the POT tool box (Flamary et al., 2021). The results are summarized in Tab. 2. As can be seen, our estimated PW discrepancies are close to the true values with the averaged relative error less than $0.2\%$, which we think is sufficiently precise for our application.

As for the efficiency, we note that the KR formulation is not advantageous in this data scale, *i.e.*, tens of points, where the computation of the exact primal form only takes less than 1 sec on a CPU, while the computation of the KR form takes a few minutes on a mid-end GPU (due to the requirement of learning the potential network). However, for large scale dataset, *e.g.*, $\sim 10^6$, which the primal form can not handle because the transport map (of size $10^6 \times 10^6$) does not even fit into the memory, the KR form can still apply (as validate in our experiments) using a GPU with 12G memory.

Finally, we stress that since the implement of the KR form depends on the underlying neural network, both the efficiency and precision naturally depend on the structure of the neural network. This is fundamentally different from the solvers of the primal form, such as the Sinkhorn algorithm or the linear program, which are fixed pipelines. Therefore, it is generally not possible to compare the efficiency and precision between the KR solvers and the primal solvers in a more rigorous way.

## F.3    MORE DETAILS OF SEC. 4.3

To estimate the gradient of the coherence energy fast, we first decompose $\mathbf{G}$ as $\mathbf{G} \approx \mathbf{Q}\Lambda\mathbf{Q}^T$ via the Nyström method (Williams & Seeger, 2000), where $k \ll r$, $\mathbf{Q} \in \mathbb{R}^{r \times k}$, and $\Lambda \in \mathbb{R}^{k \times k}$ is a diagonal

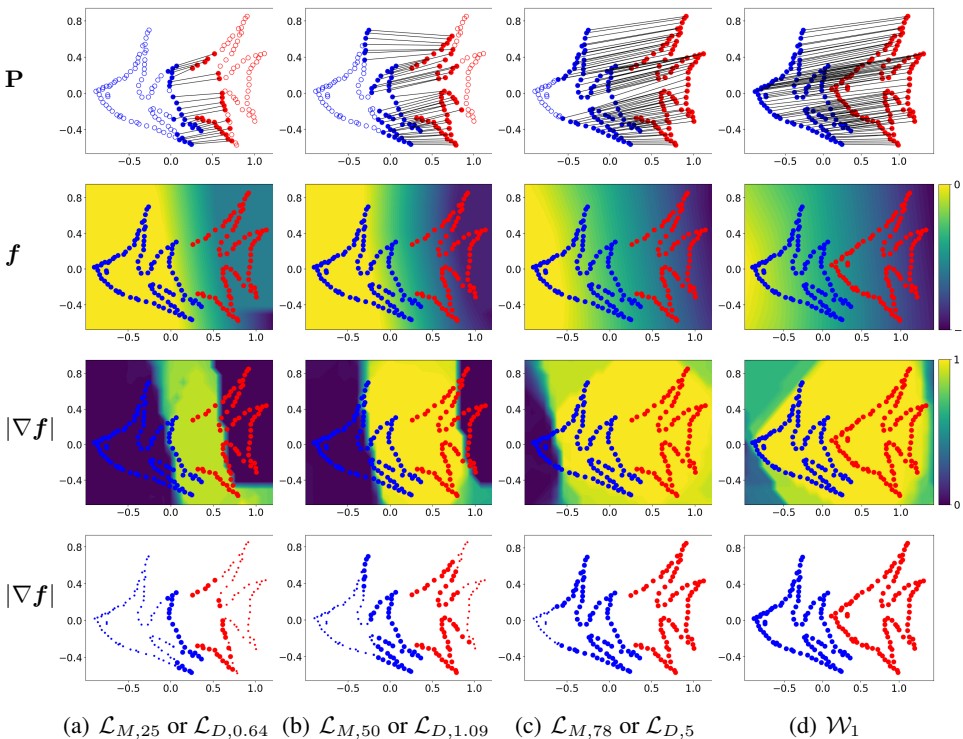

(a) $\mathcal{L}_{M,25}$ or $\mathcal{L}_{D,0.64}$ (b) $\mathcal{L}_{M,50}$ or $\mathcal{L}_{D,1.09}$ (c) $\mathcal{L}_{M,78}$ or $\mathcal{L}_{D,5}$ (d) $\mathcal{W}_1$

Figure 9: Comparison between the primal and the KR forms of the PW discrepancies on $\alpha$ (blue) and $\beta_\theta$ (red). See text for details.

Table 2: Precision of the estimated PW discrepancies for the "fish" shape in Fig. 9

|  | $\mathcal{L}_{M,25}$ | $\mathcal{L}_{M,50}$ | $\mathcal{L}_{M,78}$ | $\mathcal{L}_{D,0.648}$ | $\mathcal{L}_{D,1.09}$ | $\mathcal{L}_{D,5}$ | $\mathcal{W}_1$ |
|---|---|---|---|---|---|---|---|
| True value | 0.1354 | 0.4191 | 0.8955 | -0.0722 | -0.2795 | -4.1044 | 1.0835 |
| KR (Ours) | 0.1352 | 0.4202 | 0.8994 | -0.0724 | -0.2791 | -4.1004 | 1.0893 |

matrix. Then we apply the Woodbury identity to $(\sigma\mathcal{I} + \mathbf{Q}\Lambda\mathbf{Q}^T)^{-1}$ and obtain

$$(\sigma\mathcal{I} + \mathbf{Q}\Lambda\mathbf{Q}^T)^{-1} = \sigma^{-1}\mathcal{I} - \sigma^{-2}\mathbf{Q}(\Lambda^{-1} + \sigma^{-1}\mathbf{Q}^T\mathbf{Q})^{-1}\mathbf{Q}^T.$$

As a result, the gradient of the coherence energy can be approximated as

$$\frac{\partial \mathcal{C}(\mathcal{T}_\theta)}{\partial V} = 2\lambda(\sigma\mathcal{I} + \mathbf{G})^{-1}V \approx (2\lambda)(\sigma^{-1}V - \sigma^{-2}\mathbf{Q}(\Lambda^{-1} + \sigma^{-1}\mathbf{Q}^T\mathbf{Q})^{-1}\mathbf{Q}^TV).$$

### F.4 CONNECTIONS BETWEEN PWAN AND WGAN

In practice, the only difference between PWAN and WGAN is the structures of the potential networks. This is summarized in Tab. 3. It is easy to see that when $m_\alpha = m_\beta = m$ for $\mathcal{L}_{M,m}$, or $m_\alpha = m_\beta$ and $h > diam(\Omega)$ for $\mathcal{L}_{D,h}$, both d-PWAN and m-PWAN become WGAN.

Table 3: Comparison between the potential networks of WGAN and PWAN.

|  | input threshold | Lipschitz | negative & bounded | lower bound $h$ |
|---|---|---|---|---|
| WGAN | – | ✓ | – | – |
| d-PWAN | distance | ✓ | ✓ | fixed |
| m-PWAN | mass | ✓ | ✓ | learnable |

We note that PWAN has the following adversarial explanation. Let us call the points in $\alpha$ real points, and those in $\beta_\theta$ fake points. In the registration process, $\boldsymbol{f}_{w,h}$ is trying to discriminate real and fake

points by assigning each of them a score in range $[-h, 0]$, where real points have higher scores than fake points. $\boldsymbol{f}_{w,h}$ is so certain of a fraction of points, that it assigns the highest or lowest possible score (0 or $-h$) to them and does not change these scores easily. Meanwhile, $\mathcal{T}_\theta$ is trying to cheat $\boldsymbol{f}_{w,h}$ by moving the fraction of fake points of which $\boldsymbol{f}_{w,h}$ is not certain to obtain higher scores. In addition, $\mathcal{T}_\theta$ is also trying to move all fake points to keep the coherence energy low. The process ends when $\mathcal{T}_\theta$ cannot make further improvement on the fake points.

## F.5    EXPERIMENTAL DETAILS

**Data synthesis**    The data synthesis procedure mostly follows Hirose (2021b). Specifically, given the original point set $X \in \mathbb{R}^{r \times 3}$, the deformed point set $Y \in \mathbb{R}^{r \times 3}$ is generated by:

$$Y = X + V + \epsilon, \tag{48}$$

where $\epsilon$ is the Gaussian noise following $\mathcal{N}(0, 0.02)$, and $V \in \mathbb{R}^{r \times 3}$ is the random offset vectors. To ensure $V$ varies smoothly in the space, we sample column vector $V_j$ from a distribution $p$:

$$p(v) = \mathcal{N}(v; 0, \lambda^{-1}\mathbf{G}), \tag{49}$$

where $\mathbf{G} \in \mathbb{R}^{r \times r}$ is a Gaussian kernel defined as $\mathbf{G}_\rho(i, j) = e^{-||y_i - y_j||^2/\rho}$. To sample $V$ from ditribution $p$, we compute $V = \mathbf{G}^{\frac{1}{2}}U$, where $U \in \mathbb{R}^{r \times 3}$ is a random matrix, of which each element follows $\mathcal{N}(0, 1)$, and $\mathbf{G}^{\frac{1}{2}} \approx \mathbf{Q}\Lambda^{\frac{1}{2}}$ is computed via the Nyström method.

We use $\lambda = 10$ or $50$ and $\rho = 2$ in our experiments.

**Network structure**    The network used in our experiment is a 5-layer point-wise multi-layer perceptron with a skip connection. The detailed structure is shown in Fig. 10.

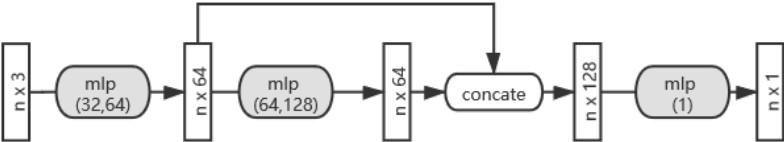

Figure 10: The structure of the network used in our experiments. The input is a matrix of shape $(n, 3)$ representing the coordinates of all points in the set, and the output is a matrix of shape $(n, 1)$ representing the potential of the corresponding points. $mlp(x)$ represents a multi-layer perceptron (mlp) with the size $x$. For example, $mlp(m, n)$ represents a mlp consisting of two layer, and the size of each layer is $m$ and $n$. We use ReLu activation function in all except the output layer. The activation function $\boldsymbol{l}(x; h) = \max\{-|x|, -h\}$ is added to the output to clip the output to interval $[-h, 0]$.

**Parameters**    We train the network $\boldsymbol{f}_{w,h}$ using the Adam optimizer (Kingma & Ba, 2014), and we train the transformation $\mathcal{T}_\theta$ using the RMSprop optimizer (Tieleman & Hinton, 2012). The learning rates of both optimizers are set to $10^{-4}$.

For experiments in Sec. 5.3, the parameters as set as follows: For PWAN, we set $(\rho, \lambda, \sigma, T) = (2, 0.01, 0.1, 2000)$. For TPS-RPM, we set $T\_finalfac = 500$, $frac = 1$, and $T\_init = 1.5$. For GMM-REG, we set $sigma = 0.5, 0.2, 0.02$, $Lambda = .1, .02, .01$, $max\_function\_evals = 50, 50, 100$ and $level = 3$. For BCPD and CPD, we set $(\beta, \lambda, w) = (2.0, 2.0, 0.1)$.

For experiments in Sec. 5.4, the parameters as set as follows: For PWAN, we set $(\rho, \lambda, \sigma, T) = (1.0, 10^{-3}, 1.0, 3000)$. For BCPD and CPD, we set $(\beta, \lambda, w) = (3.0, 20.0, 0.1)$. We also test $w = 0.4$ for these methods, but the difference is not obvious.

For experiments in Sec. F.9, the parameters as set as follows: For PWAN, we set $(\rho, \lambda, \sigma, T) = (0.5, 5 \times 10^{-4}, 1.0, 3000)$. For BCPD and CPD, we set $(\beta, \lambda, w) = (3.0, 20.0, 0.1)$.

For experiments in Sec. F.10, the parameters as set as follows: We use m-PWAN. we set $(\rho, \lambda, \sigma, T) = (1.0, 1 \times 10^{-4}, 1.0, 3000)$. For BCPD and CPD, we set $(\beta, \lambda, w) = (0.3, 2.0, 0.1)$. We also tried $w = 0.6$ but the results are similar (not shown in our experiments).

We note the parameters for CPD and BCPD are suggested in Hirose (2021a).

**Parameter selection**  The parameters $m$ in m-PWAN and $h$ in d-PWAN control the degree of alignment. Specifically, increasing $m$ or $h$ will lead to the registration of larger fraction of distributions. When $m = 0$ or $h = 0$, PWAN does not align any point (the gradients are 0 for all points); When $m = \min\{m_\alpha, m_\beta\}$ or $h = \infty$, PWAN seeks to align the largest fraction of points. Therefore, choosing overly large $m$ or $h$ may lead to the biased registration (like a DM method), while choosing overly small $m$ or $h$ may lead to the insufficient registration where too few points are aligned.

For uniform point sets, the parameters $m$ and $h$ can both be determined easily. For d-PWAN, if the averaged nearest distance between points is $s$, then $h$ can be set near $s$. Because for the aligned point sets, the points that are farther from the overlapped region than distance $s$ are likely to be outliers, and the use of $h = s$ can effectively discard those outliers. For m-PWAN, one needs to estimate the overlap ratio $\rho$ of one point set, *e.g.*, $\alpha$, then $m$ can be set as $\rho m_\alpha$. We note this is how we actually determined $m$ and $h$ in our experiments.

**Refinement**  In our experiments, we find that a nearest-point-based refinement sometimes slightly improves the performance. Specifically, when the algorithm 1 is near convergence, we can optionally run a few steps of the nearest-point-based refinement. Specifically, since algorithm 1 is near convergence, we can safely assume that the point sets are sufficiently aligned. Therefore, each point is highly likely to correspond to its nearest neighborhood in the other set within the mass or distance threshold. In other words, the correspondence matrix $\overline{\mathbf{P}} \in \mathbb{R}^{q \times r}$ can be written as

$$\overline{\mathbf{P}}_{i,j} = \begin{cases} 1 & if \ y_j = NN(x_i) \ and \ d(y_j, x_i) \leq h \\ 0 & else \end{cases} \tag{50}$$

for $\mathcal{L}_{D,h}$, and

$$\overline{\mathbf{P}}_{i,j} = \begin{cases} 1 & if \ y_j = NN(x_i) \ and \ x_i \in Nearest(m) \\ 0 & else \end{cases} \tag{51}$$

for $\mathcal{L}_{M,m}$, where $NN()$ is the nearest neighborhood function, and $Nearest(m)$ represent the nearest $m$ points in $X$ to $Y$. Thus we can slightly refine the result by switch the divergence term $\mathcal{L}$ to

$$\mathcal{L}(\alpha, \beta_\theta) = \sum_{i,j} \overline{\mathbf{P}}_{i,j} \boldsymbol{d}(x_i, \mathcal{T}_\theta(y_j))^2, \tag{52}$$

and update a few steps of the transformation $\mathcal{T}_\theta$ via gradient descent.

## F.6  MORE DETAILS OF SEC. 5.2

The KL divergence and the $L_2$ distance between point set $X$ and $Y$ are formally defined as

$$L_2(X, Y) = \sum_{\substack{x_i, x_j \in X \\ y_i, y_j \in Y}} \frac{1}{q^2} \phi(0|x_i - x_j, 2\sigma) + \frac{1}{r^2} \phi(0|y_i - y_j, 2\sigma) - \frac{2}{qr} \phi(0|x_i - y_j, 2\sigma), \tag{53}$$

$$KL(X, Y) = -\frac{1}{q} \sum_{y_j \in Y} \log\left(\omega \frac{1}{q} + (1 - \omega) \sum_{x_i \in X} \frac{1}{r} \phi(y_j|x_i, \sigma)\right), \tag{54}$$

where $\phi(\cdot|u, \sigma)$ is the Gaussian distribution with mean $u$ and variance $\sigma$. For simplicity, we set $\sigma = 1$ and $\omega = 0.2$ for KL and $L_2$.

## F.7  MORE DETAILS OF SEC. 5.3

We present more results of the experiments with $N = 2000$ in Fig. 11 and Fig. 12, and the corresponding quantitative results are shown in Tab. 4 and Tab. 5. We do not show the results of TPS-RPM on the second experiment, as it generally fails to converge. As can be seen, PWAN successfully registers the point sets in all cases, while all baseline methods bias toward to the noise points or to the non-overlapped region when outlier ratio is high, except for TPS-RPM which shows strong robustness against noise points comparable with PWAN in the first example.

To obtain a sense of the learned potential network $\boldsymbol{f}_{w,h}$, we visualize the potential at the end of the registration process. The results are shown in Fig. 13 and Fig. 14. We represent the value of potential and the gradient norm at each point by colors, where brighter color indicates higher value. As can be seen in Fig. 14(c) and 13(c), the network assigns higher values to the points in $\alpha$ while assigning lower values to the points in $\beta_\theta$. In addition, as shown in Fig. 14(d) and 13(d), most of outliers, including noise points and the points in non-overlapping region, have low gradients, *i.e.*, they are successfully discarded by the network.

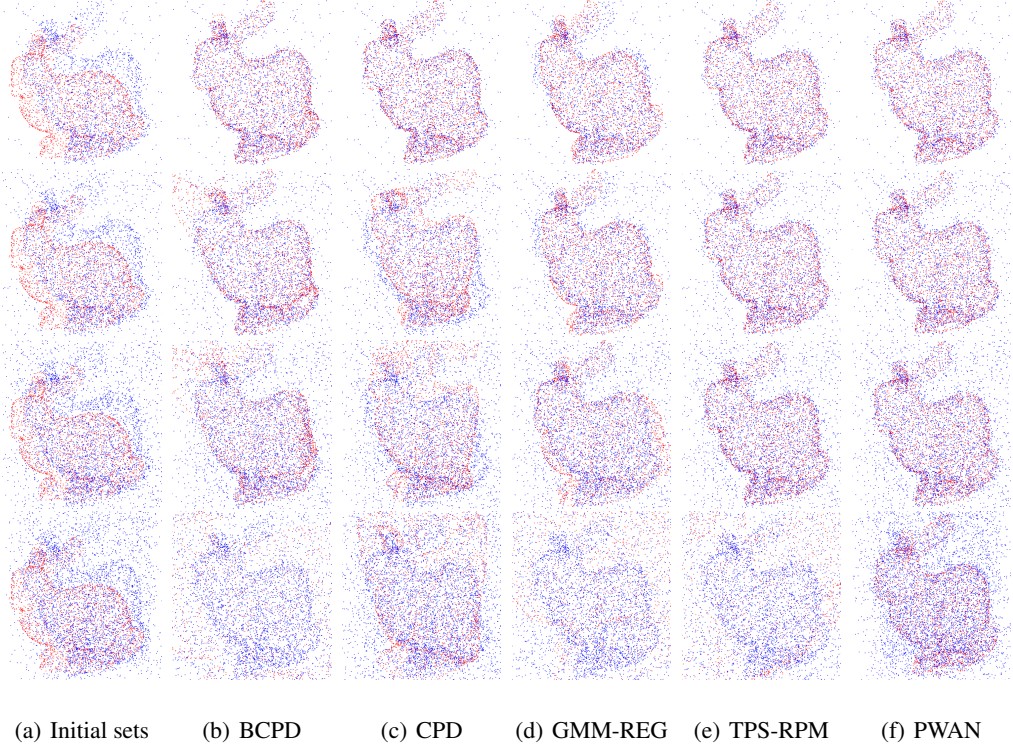

| (a) Initial sets | (b) BCPD | (c) CPD | (d) GMM-REG | (e) TPS-RPM | (f) PWAN |

Figure 11: An example of registering noisy point sets. The outlier/non-outlier ratios of the point sets shown here are 0.2 (1st row), 0.6 (2nd row), 1.2 (3rd row) and 2.0 (4th row).

Table 4: Quantitative comparison of the registration results shown in Fig. 11 using MSE.

| Outlier/Non-outlier ratio | BCPD | CPD | GMM-REG | TPS-RPM | PWAN |
|---|---|---|---|---|---|
| 0.2 | 0.011 | **0.0015** | 0.018 | 0.0032 | 0.0031 |
| 0.6 | 0.075 | 0.083 | 0.022 | **0.0026** | 0.0030 |
| 1.2 | 0.12 | 0.21 | 0.042 | 0.0033 | **0.0030** |
| 2.0 | 1.24 | 0.289 | 3.76 | 2.354 | **0.004** |

Table 5: Quantitative comparison of the registration results shown in Fig. 12 using MSE.

| Overlap ratio | BCPD | CPD | GMM-REG | d-PWAN | m-PWAN |
|---|---|---|---|---|---|
| 0.57 | 0.18 | 0.92 | 0.32 | 0.017 | **0.015** |
| 0.75 | 0.028 | 0.45 | 0.043 | **0.0044** | 0.0090 |
| 1 | **0.00072** | 0.0025 | 0.0078 | 0.0037 | 0.0038 |

## F.8    MORE DETAILS OF SEC. 5.4

We provide more details regarding the computation time of our method. We first sample $q = r$ points from the bunny shape, where $q = 10^5$ or $7 \times 10^5$. We then run both the 1 GPU version and the 2 GPU version of PWAN 100 steps, and report their computation time in Fig. 15, where $q$-$M$GPU represents registering $q$ points using $M$ GPU. As can be seen, the majority of time is spent

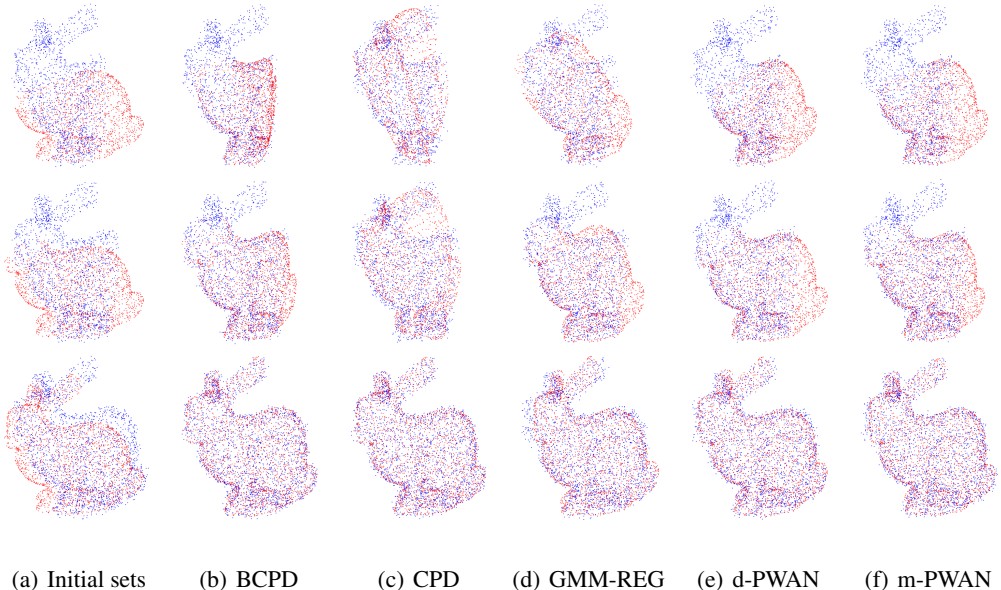

(a) Initial sets     (b) BCPD     (c) CPD     (d) GMM-REG     (e) d-PWAN     (f) m-PWAN

Figure 12: An example of registering partially overlapped point sets. The overlap ratio of the point sets shown here are $0.57$ (1st row), $0.75$ (2nd row) and $1$ (3rd row).

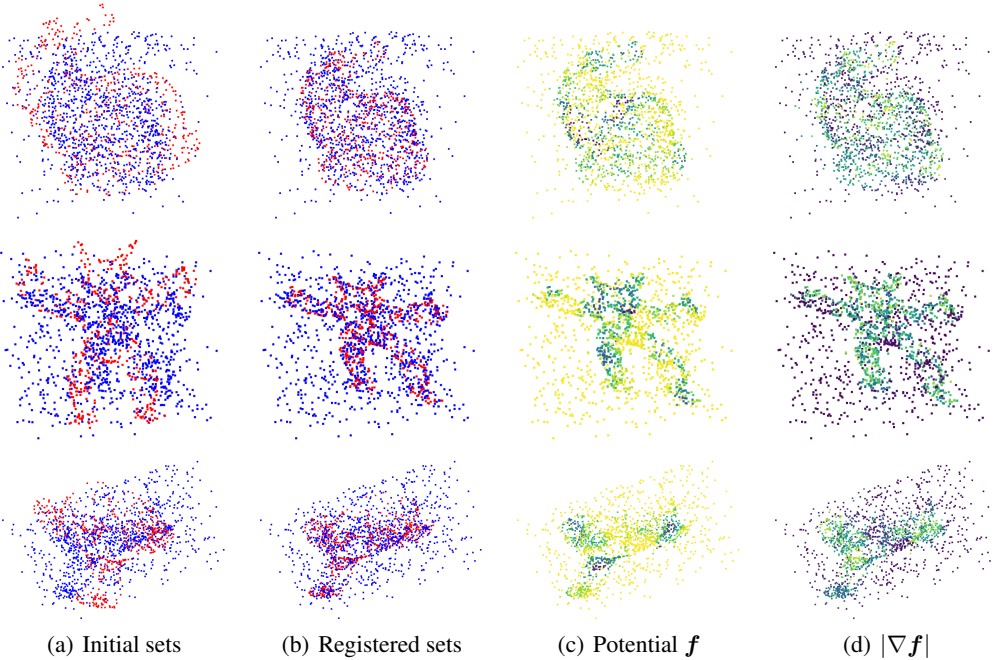

(a) Initial sets       (b) Registered sets       (c) Potential $f$       (d) $|\nabla f|$

Figure 13: Visualization of the learned potentials on point sets with extra noise points.

on updating the network, and the parallel training can effectively reduce the computation time for updating the network. In addition, the gain of speed increase as the number of points, *i.e.*, we get $1.3\times$ speedup when $q = 10^5$, while the speedup is $1.9\times$ when $q = 7 \times 10^5$, which is close to the theoretical speedup value $2$.

Finally, we evaluate PWAN on large scale point sets. We generate noisy and partially overlapped armadillo datasets as stated in Sec. 5.3. We compare PWAN with BCPD and CPD, because they are the only baseline methods that are scalable in this experiment. We present some registration results in

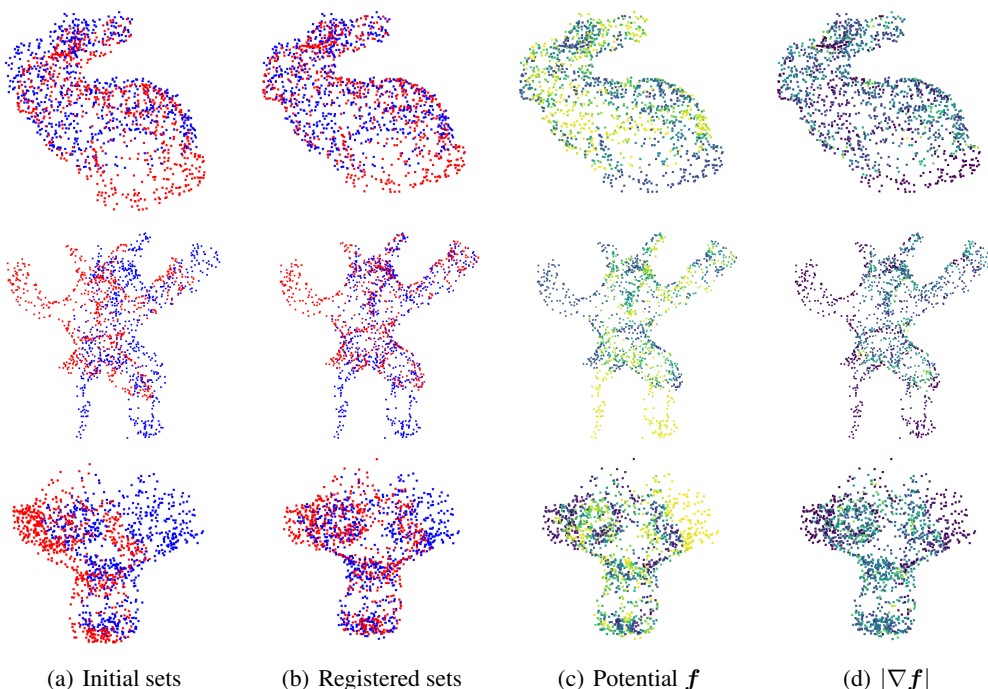

(a) Initial sets          (b) Registered sets          (c) Potential $f$          (d) $|\nabla f|$

Figure 14: Visualization of the learned potentials on partially overlapped point sets.

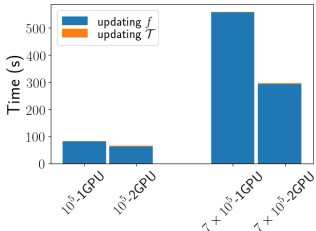

Figure 15: Computation time of our method.

Fig. 16. As can be seen, our method can handle both cases successfully, while both CPD and BCPD bias toward outliers.

More training details of PWAN is shown in Fig. 17, Fig. 18 and Fig. 19. As can be seen in Fig. 17(a), 18(a) and 19(a), the source sets are matched to the reference sets smoothly. Fig. 17(b), 18(b) and 19(b) suggest that at the end of the registration process, most of outliers are correctly discarded. Besides, Fig. 17(c), 18(c) and 19(c) implies that the norm of gradient of the network is indeed controlled below 1, *i.e.*, it is indeed Lipschitz. In addition, both the training loss and the MSE decrease smoothly in the training process.

### F.9 EXPERIMENT ON THE HUMAN FACE DATASET

We evaluate our method in the space-time faces dataset (Zhang et al., 2008), which consists of a time series of point sets sampled from a real human face. Each face consists of $23,728$ points and the true correspondence between faces are known. We use the faces at time $i$ and $i + 20$ as the source and the reference set, where $i = 1, ..., 20$. All point sets in this dataset are the same size and are completely overlapped.

The registration results are shown in Tab. 6, where we can see PWAN outperforms both CPD and BCPD. We present the examples of the registration results in Fig. 20. As can be seen, PWAN successfully aligns the faces in different time points.

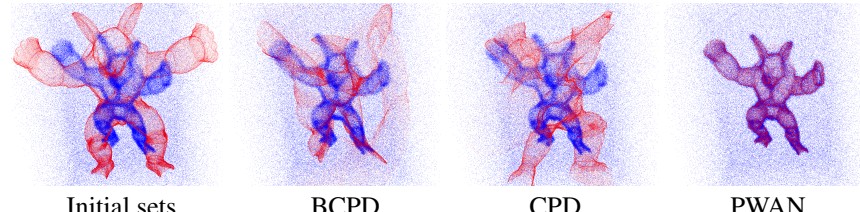

(a) An example of registering noisy point sets. The source and refernece sets contain $8 \times 10^4$ and $1.76 \times 10^5$ points respectively.

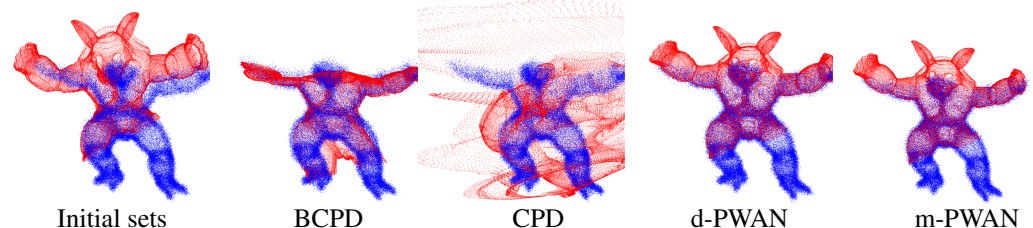

(b) An example of registering partially overlapped point sets. The source and refernece sets both contain $7 \times 10^4$ points.

Figure 16: Examples of registering large scale point sets.

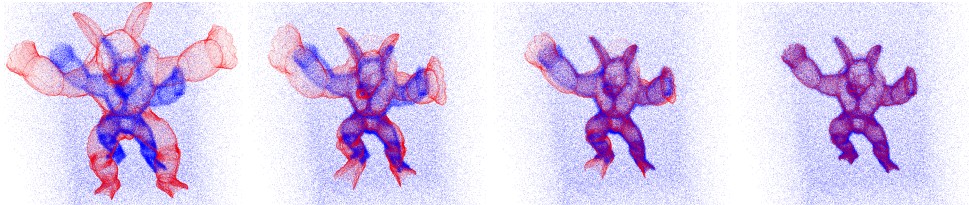

(a) Registration trajectory. The registration process proceedes from left to right.

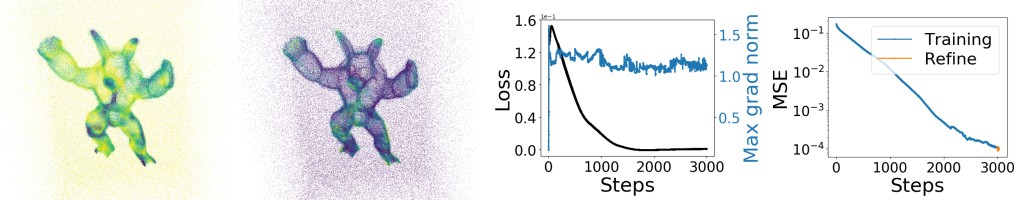

(b) Visualization of the potential $f$ (left) and it gradient $|\nabla f|$ (right) at the final step.

(c) Statistics of the training process.

Figure 17: One example of registering the noisy "armadillo" datasets. The source and refernece sets contain $8 \times 10^4$ and $1.76 \times 10^5$ points respectively.

Table 6: Registration results of the space-time faces dataset.

| BCPD | CPD | PWAN |
|---|---|---|
| $0.0017 \pm 0.001$ | $0.00049 \pm 0.0002$ | $0.00032 \pm 0.000089$ |

## F.10   EXPERIMENT ON THE 3D HUMAN DATASET

We evaluate our method on a challenging human shape dataset (DataSet), which is taken from a SHREC'19 track called "matching humans with different connectivity". This dataset consists of $44$ shapes, and we manually select 3 pairs of shapes for our experiments. To generate a point set of each shape, we first sample 50000 random points from the surface of the 3D mesh, and then apply voxel grid filtering to down-sample the point set to less than $10000$ points. The description for the selected point sets is presented in Tab. 7

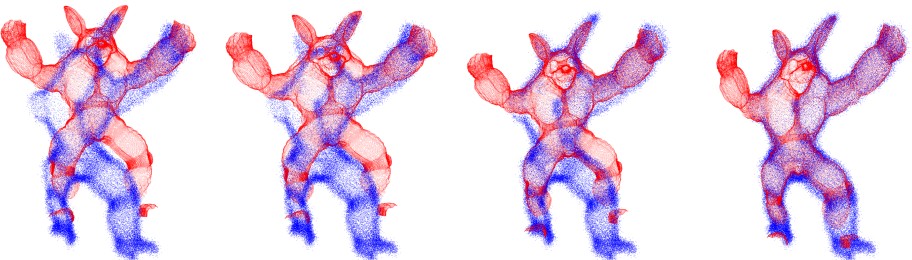

(a) Registration trajectory. The registration process proceedes from left to right.

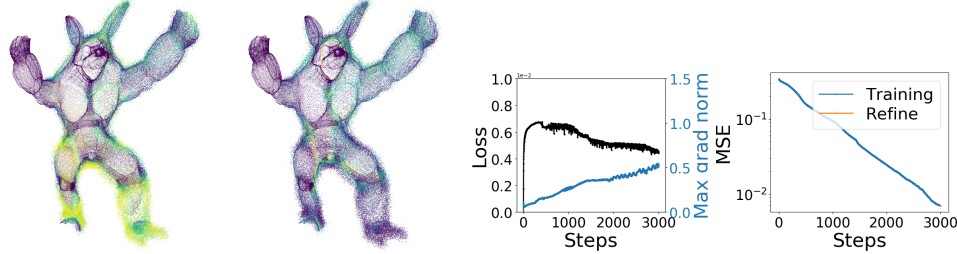

(b) Visualization of the potential $\boldsymbol{f}$ (left) and it gradient $|\nabla \boldsymbol{f}|$ (right) at the final step.

(c) Statistics of the training process.

Figure 18: One example of registering the partially overlapped "armadillo" datasets using d-PWAN. Each point set consists of $7 \times 10^4$ points.

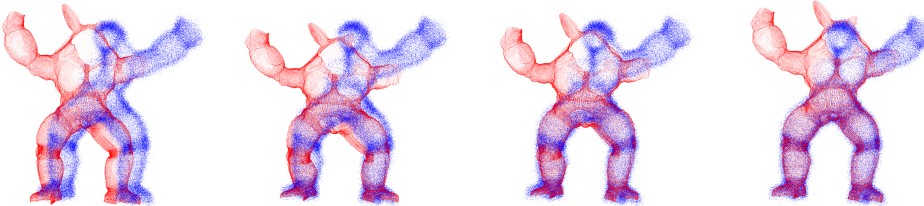

(a) Registration trajectory. The registration process proceedes from left to right.

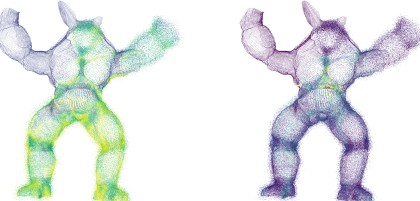

(b) Visualization of the potential $\boldsymbol{f}$ (left) and it gradient $|\nabla \boldsymbol{f}|$ (right) at the final step.

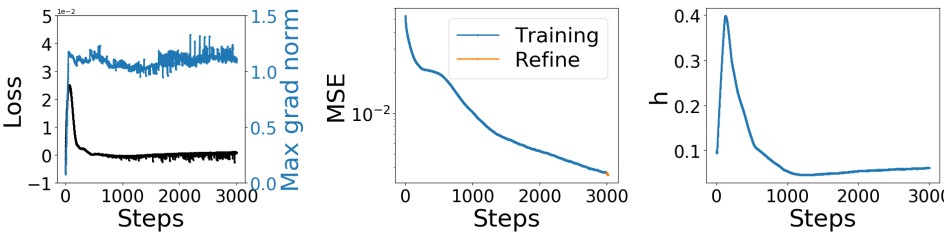

(c) Statistics of the training process.

Figure 19: One example of registering the partially overlapped "armadillo" dataset using m-PWAN. Each point set consists of $7 \times 10^4$ points.

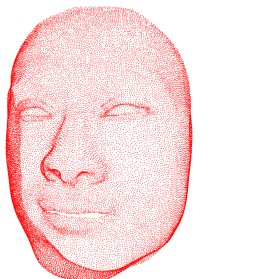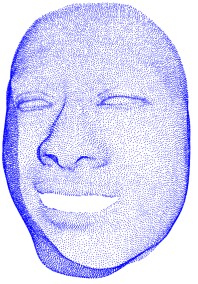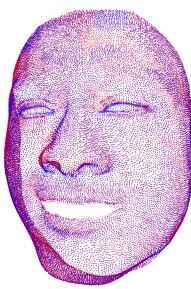

(a) The aligned point sets (right) is obtained by matching the 1-st frame (left) to the 21-st frame (middle)

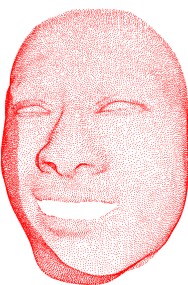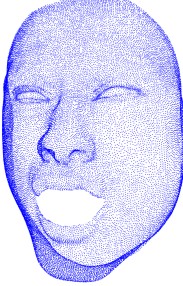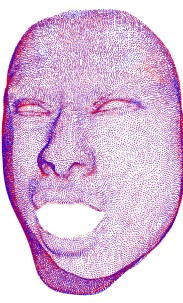

(b) The aligned point sets (right) is obtained by matching the 19-st frame (left) to the 39-st frame (middle)

Figure 20: Examples of our registration results on the human faces dataset. Zoom in to see the details.

Table 7: Point sets used for registration. no.$m$ represents the m-th shape in the dataset (DataSet).

|  | (no.1, no.42) | (no.18, no.19) | (no.30, no.31) |
|---|---|---|---|
| Size | (5575, 5793) | (6090, 6175) | (6895, 6792) |
| Description | same pose different person | different pose same person | different pose different person |

We conduct the following two experiments. In the first experiment, we evaluate our method on registering the complete point sets. In the second experiment, we evaluate our method on the more challenging partial matching problem, where we generate incomplete point sets by manually cropping a fraction of the no.30 and no.31 point sets. We consider 3 types of partial matching problem, *i.e.*, match incomplete set to complete set, match complete set to incomplete set and match incomplete set to incomplete set. For both of these two experiments, we compare our method with CPD and BCPD, and we only present qualitative registration results, because we do not know the true correspondence between point sets.

The results of the first experiment is shown in Fig. 21. As can be seen, PWAN can handle both the local deformations (1-st row) and the articulated deformations (2-nd and 3-rd rows) well, and it produces good full-body registration results. In contrast, although CPD and BCPD can handle local deformations relatively well (1-st row), they have difficulties aligning point sets with large articulated deformations, as significant registration errors are observed near the limbs (2-nd and 3-rd rows).

The results of the second experiment is shown in Fig. 22. As can be seen, both CPD and BCPD fail in this experiment, as the non-overlapping points are seriously biased. For example, in the 3-rd row, they both wrongly match the left arm to the body, which causes highly unnatural artifacts. In contrast, the proposed PWAN can handle the partial matching problem well, since it successfully maintains the shape of non-overlapping regions, which contributions to the natural registration results.

Finally, we note that although the proposed PWAN generally produces reasonable full-body registration results, it has some difficulties handling the local details. For example, the hands in Fig. 21 and 22 are generally not well aligned. This drawback might be alleviated by considering local constraints such as Ge et al. (2014) in the future. In addition, it is worth noticing that the aligned point sets in the second experiments are natural and do not exist in the original dataset. These results suggests the potential of our method in other practical tasks such as point set completion and point sets merging.

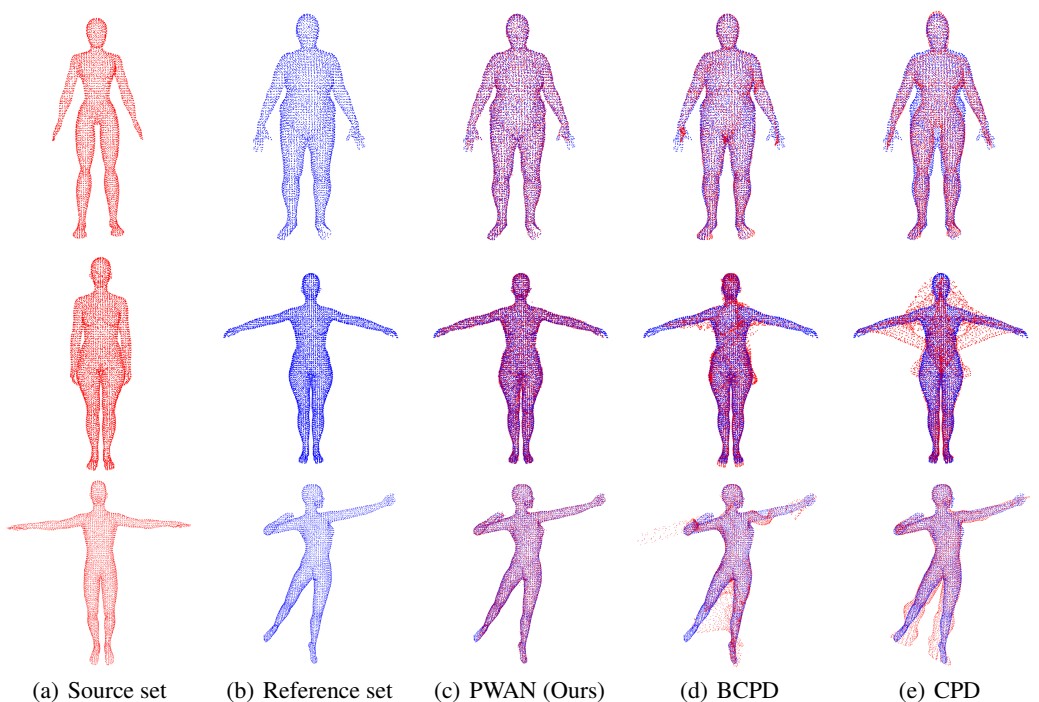

|  (a) Source set | (b) Reference set | (c) PWAN (Ours) | (d) BCPD | (e) CPD |

Figure 21: The results of registering complete point sets no.1 to no.42 (1-st row), no.18 to no.19 (2-nd row), and no.30 to no.31 (3-rd row). Our results are compared against BCPD (Hirose, 2021a) and CPD (Myronenko & Song, 2010). Zoom in to see the details.

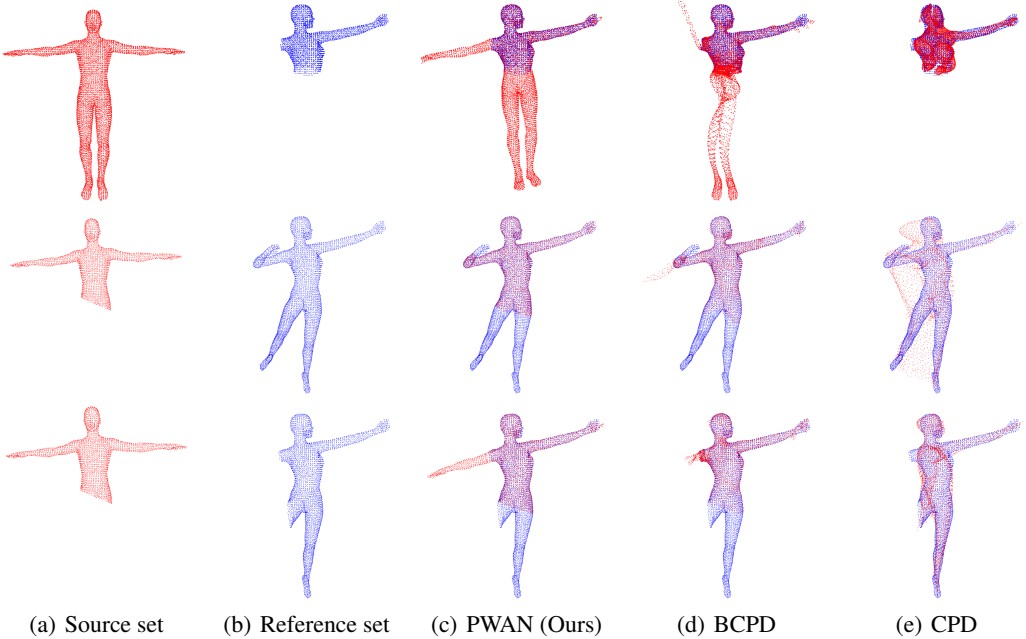

|  (a) Source set | (b) Reference set | (c) PWAN (Ours) | (d) BCPD | (e) CPD |

Figure 22: Registering incomplete point sets no.30 to no.31. We present the results of the complete to incomplete (1-st row), incomplete to complete (2-nd row) and incomplete to incomplete (3-rd row) registration. Our results are compared against BCPD (Hirose, 2021a) and CPD (Myronenko & Song, 2010). Zoom in to see the details.

