# OpenReview forum: "Partial Wasserstein Adversarial Network for Non-rigid Point Set Registration"
_ICLR.cc/2022/Conference — ICLR 2022 Poster_

### Official Review · Reviewer_dwHk · 2021-10-28

**Correctness:** 3
**Technical Novelty And Significance:** 3
**Empirical Novelty And Significance:** 3
**Recommendation:** 6
**Confidence:** 3

**Main Review:**

Strengths:

1) It's very nice to have a method to avoid the computation of dense matching matrix in an optimal transport framework;
2) The mathematical derivations are extensive and seem to be correct.

Weaknesses:

I would say the experiments are only validated on very small-scale synthetic and real-world datasets. I would expect the following experiments:

1) Experiments on the 3D Human dataset [Section 4.3, Non-rigid Point Set Registration with Global-Local Topology Preservation]

2) For synthetic experiments, it would be better to use the modelnet40 dataset, which contains many different shapes. Only using three shapes [bunny, armadillo and monkey] to train and validate methods has the risk of over-fitting.

3) For the limited real-world experiment on the face dataset [E.10], it would be better to give more details. For example, what's the overlapping ratio between pairwise point clouds? How does the proposed method perform under different overlapping ratios?

4) It's unclear whether the proposed method would be open-sourced. Since this paper contains many equations, re-implementing it would be very hard.

**Summary Of The Paper:**

This paper proposes a partial Wasserstein adversarial network to register two non-rigid point clouds.

The major contribution is deriving a dual optimization objective to avoid the computation of a dense matching matrix.

Experiments on toy/synthetic datasets and one small-scale real-world face dataset show the effectiveness of the proposed method.

**Summary Of The Review:**

Overall, It's very nice to have a method to avoid the computation of dense matching matrix in an optimal transport framework. However, the limited experiments have not convinced me of its effectiveness on real-world datasets.

---

> ### Author Response · Authors · 2021-11-19
> **Response to reviewer dwHk**
>
> We thank the reviewer for the valuable comments. We address the concerns below.
>
> 1. **''..Experiments on the 3D Human dataset..''**
>
> We have added a new experiment on 3D human dataset in Appx.F.10, where we show that our method can handle complicated deformations and incomplete data.
>
> 2. **''....it would be better to use the modelnet40 dataset.... methods has the risk of over-fitting.''**
>
> The proposed method dose not suffer from over-fitting.
> Because our method is unsupervised, i.e.,
> our method directly applies to each pair of point sets and the true registrations are not given in the registration process,
> thus it is not possible to overfit.
>
> 3. **''...face dataset [E.10],
> it would be better to give more details.
> For example,
> what's the overlapping ratio between pairwise point clouds?''**
>
> In the face experiments,
> the point sets are always fully overlapped.
> We have added this sentence to Appx E.10.
> >All point sets in this dataset are the same size and are completely
> overlapped.
>
> 4. **''It's unclear whether the proposed method would be open-sourced. ''**
>
> Yes, we will release our source code.

---

> > ### Author Response · Authors · 2021-12-01
> > **Please let us know whether we have addressed your concerns.**
> >
> > We thank the reviewer for the valuable suggestions.
> > According to the comments,
> > we have added a new experiment on the 3D human shape,
> > and we have responded to the other concerns.
> > Please let us know whether the new experiment is convincing,
> > and whether our rebuttal has addressed your concerns.

---

### Official Review · Reviewer_U9Y7 · 2021-11-03

**Correctness:** 4
**Technical Novelty And Significance:** 3
**Empirical Novelty And Significance:** 2
**Recommendation:** 6
**Confidence:** 4

**Main Review:**

As to the strengths of the paper, the authors are able to make a novel contribution to a classic problem. It is pleasing that the point sets are modelled as discrete distributions without the need of making smoothing like GMM based methods for instance do. The theoretical part of deriving the KR duality for the two types of chosen discrepancies is clearly a strength of the paper even though it is a harder part of the paper to read.

As to the weaknesses, the paper is generally relatively easy to read, however the toughest part in math could be improved for clarity to help the reader with the intuition what is going on. Especially the part related to the adversarial learning and the approximation of PW discrepancy could be described better. The experimental evaluation is performed with rather simplistic datasets for which more realistic data set could be used to improve the presentation.

**Summary Of The Paper:**

The paper proposes a method for registration of two point sets by formulating the problem as partial distribution matching problem. The authors utilise the partial Wasserstein (PW) discrepancy, and show how its gradient can be computed. They further propose a partial Wasserstein adversarial network that approximates the PW discrepancy. The authors evaluate the method against four method against two artificial datasets in which the method shows good results.

**Summary Of The Review:**

In summary, the work can be seen as well founded paper with clear theoretic contribution equipped with a practical method based on neural formulation of the registration problem. Good results on the selected datasets, though an evaluation with a more complicated datasets would improve the presentation.

---

> ### Author Response · Authors · 2021-11-19
> **Response to reviewer U9Y7**
>
> We thank the reviewer for the valuable comments.
>
> To make our our paper more readable,
> we have made the following modifications:
> 1. We have revised Appx.F.2 to provide an example for the KR approximations.
> 2. For the adversarial learning part, we have revised the last paragraph of Sec.4.4 to provide an example for registering 1D examples.
>
> Besides,
> we have added a new experiment on 3D human dataset in Appx.F.10,
> where we show that our method can handle complicated deformations and incomplete data.

---

> ### Author Response · Authors · 2021-12-01
> **Please let us know whether we have addressed your concerns.**
>
> We thank the reviewer for the valuable suggestions.
>
> According to the comments,
> we have provided concrete examples for the KR formulations and the adversarial learning,
> and we have added a new experiments on the challenging 3D human shape dataset.
> Please let us know whether these modifications increase the readability of our paper,
> and whether the new experiment demonstrates the effectiveness of our method on complicated datasets.

---

### Official Review · Reviewer_y2JC · 2021-11-03

**Correctness:** 4
**Technical Novelty And Significance:** 2
**Empirical Novelty And Significance:** 3
**Recommendation:** 6
**Confidence:** 3

**Main Review:**

The paper is interesting and deals with an important problem: how to deal with outliers when registering 3d shapes.
The transformation $\mathcal{T}_\theta$ is found by simultaneously find a global transformation and optimizing a partial optimal transportation distance.
I have few concerns that should be clarified:
- eq. (2) is a reformulation of eq. (1). If there is a reason to have the two formulations, it should be better highlighted. Otherwise, it would add some clarity to avoid considering equivalent formuations.
- the dual of the partial optimal transport problem is described in [1]. If there is some novelty here, it should be emphasized. Otherwise, appropriate references should be cited.
- the choice of the parameters $m$/$h$ is the key of the success of the method. It would be interesting to add a discussion about it. What happens if their values are under/over estimated? Is there any thumb rule to choose their values?
- Section 5.2 is unclear. Is there any reason that the translated dataset should not match the outliers when t is large? What prevent PWAN to align dataset Y to outliers?


[1] Caffarelli, L. A., & McCann, R. J. (2010). Free boundaries in optimal transport and Monge-Ampere obstacle problems. Annals of mathematics, 673-730.

**Summary Of The Paper:**

Paper proposes PWAN, a method that can be seen as an extension of WGAN in the presence of noise.
It uses the partial formulation of the wasserstein metric, allowing discarding points that are noise or outliers. It relies on the dual formulation of the problem to derive an efficient algorithm to learn a transformation that maps the target points toward the source points. Extensive experiments on toy datasets and 3 point clouds show that the method performs well, even in the presence of noise.

**Summary Of The Review:**

An interesting paper deals with an important problem but lacks some appropriate references.

---

> ### Author Response · Authors · 2021-11-19
> **Response to reviewer y2JC**
>
> We thank the reviewer for the valuable comments. We address the concerns below.
> 1. **''eq. (2) is a reformulation of eq. (1)..... it would add some clarity to avoid considering equivalent formuations''**
>
> Thanks for pointing this out.
> They are related formulations but they are not equivalent objective functions.
> We have revised the last paragraph in the preliminary section to clarify this:
> >In this work,
> we use $\\mathcal{L}\_{D,h}$ and $\\mathcal{L}\_{M,m}$ as two types of objective functions to partially match the distributions.
> Note these two objective functions are closely related.
> Because according to the Lagrange duality,
> for a fixed $(\\alpha, \\beta, m)$,
> there exists a $h^* \\geq 0$,
> such that the solution to $\\mathcal{L}\_{D,h^*}(\\alpha, \\beta)$ recovers to that to $\\mathcal{L}\_{M,m}(\\alpha, \\beta)$.
> Nevertheless,
> these two objective functions are not equivalent.
> Because when $\\alpha$ or $\\beta$ varies during iterations,
> the corresponding $h^*$ varies accordingly.
> In other words,
> minimizing $\\mathcal{L}\_{M,m}$ is generally not equivalent to minimizing $\\mathcal{L}\_{D,h}$ with any fixed $h$.
>
> 2. **''The dual of the partial optimal transport problem is described in [1]. If there is some novelty here, it should be emphasized....''**
>
> We apologize for the confusion.
> The Kantorovich duality considered in Corollary 2.7 in Caffarelli \text{\&} McCann, 2010 (called ``the dual form'' in our paper) and the Kantorovich-Rubinstein (KR) duality discussed in our paper are two different formulations.
> This can be seen in Table 1 in the appendix,
> where the dual form is listed in the 2nd row 2nd column,
> and our main focus is the KR forms in the 3rd row.
> The KR forms are special cases of the dual form when the cost function is the distance ($c(x,y)=d(x,y)$).
> More discussions amongst these formulations are given in Appx.C.2.
>
> We have added a reference to Tab.1 in the paragraph after Theorem 2:
> >See Tab. 1 in the appendix for a comparison of equivalent formulations
>
> 3. **''The choice of the parameters m/h...What happens if their values are under/over estimated? Is there any thumb rule to choose their values?''**
>
> Roughly speaking,
> increasing $m$ or $h$ will lead to the registration of larger fraction of distributions.
> Therefore,
> choosing overly large $m$ or $h$ may lead to the biased registration like a DM method,
> while choosing overly small $m$ or $h$ may lead to the insufficient registration where too few points are aligned.
>
> For uniform point sets,
> the parameters $m$ and $h$ can both be determined easily.
> For d-PWAN,
> if the averaged nearest distance between points is $s$,
> then $h$ can be set near $s$.
> Because for the aligned point sets,
> the points that are farther from the overlapped region than distance $s$ are likely to be outliers.
> For m-PWAN,
> one needs to estimate the overlap ratio $\rho$ of one point set,e.g, $\alpha$,
> then $m$ can be set as $\rho m_\alpha$.
> This is how we determined $m$ and $h$ in our experiments.
>
> We have added a new paragraph in Appx.F.5 to briefly discuss it.
>
> 4. **''...Is there any reason that the translated dataset should not match the outliers when t is large?
> What prevent PWAN to align dataset Y to outliers?''**
>
> In this toy example,
> the correct alignment t=0 is the perfect alignment,
> and the biased alignment t=6.5 is a bad one.
> Because when t=0, each point in Y is exactly matched to a different point in X,
> i.e.,
> Y is perfectly matched.
> But when t=6.5, at most two innermost points in $Y$ is matched to the points in $X$.
> Therefore,
> we always wish to find t=0 no matter what the current $t$ and $N$ are.
>
> It is possible for PWAN to align Y to outliers,
> i.e., it may converge to t=6.5,
> but it is more likely to converge to the correct alignment t=0.
> Specifically,
> as shown in Fig.6,
> the two PW discrepancies indeed have local minimum t=6.5 when $N$ is large,
> which means they might be trapped in t=6.5.
> However,
> they always have deeper local minimum at t=0,
> which suggests that they are more likely to converge to the correct alignment t=0.
>
> This is because for PW discrepancies considered here ($\\mathcal{L}\_{D,2}$ and $\\mathcal{L}\_{M,10}$),
> the outliers actually have no influence on the correct alignment t=0,
> i.e,
> unlike KL and $L_2$ divergence,
> **the local minimum t=0 of PW discrepancies will not becomes shallower when N increases.**
> Because according to their definitions,
> the PW discrepancies only depend on the points within the mass or distance threshold,
> and for the correct alignment t=0,
> the outliers are actually beyond the threshold.
> Specifically,
> when t=0,
> for $\\mathcal{L}\_{D,2}$, the distance between $Y$ and the outliers are farther than threshold $h=2$,
> and for $\\mathcal{L}\_{M,10}$, the outliers are not the nearest $m=10$ points to Y.
>
> We have revised the last paragraph in Sec.5.2 to make this clearer.

---

> ### Author Response · Authors · 2021-12-01
> **Please let us know whether you have further concerns.**
>
> We thank the reviewer for the time reviewing our paper.
>
> We would like to know whether the discussions have addressed all the concerns.

---

### Official Review · Reviewer_7kLR · 2021-11-05

**Correctness:** 4
**Technical Novelty And Significance:** 3
**Empirical Novelty And Significance:** 3
**Recommendation:** 6
**Confidence:** 4

**Details Of Ethics Concerns:**

I do not believe this paper involve any ethics issue.

**Main Review:**

Strengths:

+ The paper theoretically derives the Kantorovich–Rubinstein duality for two types of the PW discrepancy, i.e., the distance-type and the mass-type discrepancy

+ The paper propose PWAN, for large scale PDM problem.

+ Experimental results on the point set registration task prove that the proposed PWAN is robust, scalable and performs more favorably
than the state-of-the-art methods

Weaknesses:

-  The paper is rather dense, which may to some extent not well self-contained.

-  The paper only evaluated on the point set registration task, other related task such as 2D shape matching could be exploited.

**Summary Of The Paper:**

This paper deals with the problem of non-rigid point set registration, where a method for large scale partial distribution matching (PDM) problem is proposed by utilizing the partial Wasserstein-1 (PW) discrepancy.  The paper theoretically derive the Kantorovich–Rubinstein duality for the PW discrepancy, and show its gradient can be explicitly computed. Based on these theoretical results, the paper proposed a partial Wasserstein adversaria network (PWAN), which approximates the PW discrepancy by a neural network, and learns the transformation adversarially with the network. Experimental results on point set registration tasks such that the proposed PWAN is robust, scalable and performs more favorably than the state-of-the-art methods.

**Summary Of The Review:**

The theoretical contributions are justified with experimental evaluation on the point set registration task. The experimental evaluations could be more extensive, for example, other related tasks.

---

> ### Author Response · Authors · 2021-11-19
> **Response to reviewer 7kLR**
>
> We thank the reviewer for the valuable comments. We address the concerns below.
> 1. **''The paper is rather dense, which may to some extent not well self-contained.''**
>
> We have added a new section Appx.B to introduce other related optimal transport algorithms.
> We hope these materials can be useful for the readers who wish to find related materials not included in our work.
>
> 2. **''The paper only evaluated on the point set registration task, other related task such as 2D shape matching could be exploited.''**
>
> We have added a new experiment on matching 3D human shape in Appx.F.10,
> which is more challenging than 2D shapes.
> We hope this experiment can demonstrate the ability of our method in handling practical point sets.

---

> ### Author Response · Authors · 2021-12-01
> **Please let us know whether we have addressed your concerns.**
>
> We thank the reviewer for the valuable suggestions.
>
> According to the comments,
> we have added a new experiments on the challenging human shape registration,
> and we have added a new section Appx.B discussing the connections between our work and other related work.
>
> Please let us know whether these modifications have addressed your concerns.

---

### Official Review · Reviewer_x4db · 2021-11-06

**Correctness:** 3
**Technical Novelty And Significance:** 2
**Empirical Novelty And Significance:** 3
**Recommendation:** 6
**Confidence:** 3

**Main Review:**

 ***Strengths***
- The paper  considers point set registration problem consisting in
 in learning a non-rigid transformation that matches a transformed
source set to a reference one while avoiding noisy points or outliers that do have
correspondences in the other set.  To ensure the robustness of the
matching a partial distribution matching is sought using partial
optimal transport.  Specifically the partial Wasserstein-1 (PW), where only a portion of the probability mass is moved,  is
considered.

- Two equivalent formulations are proposed, the plain PW problem
  coined $\mathcal{L}_{M,m}$ with explicitly mass constraint and the

relaxed version $\mathcal{L}_{D,h}$. The main results of the paper
  are the derivation of the Kantorovitch-Rubinstein dual of both
  problems. The dual is a constrained optimization problem
  that involves a potential function defined as a neural network (NN). By
  relaxation of the Lipschitz constraint on the NN, the dual  is amenable to a
gradient framework and hence, to a large scale computation.

- Learning of $\mathcal{T}$ by minimizing the PW discrepancy with a
coherence regularization  leads to a min-max problem similar to  the
one of the Wasserstein GAN. The overall optimization algorithm is
given.

***Weaknesses***
- As stated in the appendix, the Kantorovitch-Rubinstein (KR)  dual of  $\mathcal{L}_{D,h}$ straightforwardly
derives from (Lellmann et al., 2014; Schmitzer & Wirth, 2019).
By convexity argument, there exist $h \geq 0$ such that problem

$\mathcal{L}_{D,h}$ yields to a solution of problem

$\mathcal{L}_{M,m}.$

Hence, the dual Kantorovitch-Rubinstein  form of
$\mathcal{L}_{M,m}$ should follow. In that regard, the KR dual
derivations of both PW problems appear issued directly from the
aforementioned works. This renders the main theoretical results of the
paper rather incremental.

- The relaxation  of the Lipschitz constraint on the NN implies that
  the obtained dual objective function (even if the dual problem is
  optimized to fully convergence) no longer represents the PW
  discrepancy but an approximation. Hence, the gradient computation is biased and results in the loss of theoretical guarantees on the
retrieved solution $\mathcal{T}$ of the point set registration problem (3).

- The paper lacks theoretical analysis of the convergence rate, the time
  complexity of the PW computation.  In the same
  vein, it lacks the comparison of the primal and dual solutions
  (computation time, precision  of the PW losses) or the empirical convergence illustration of Algorithm 1.

- It is not explicitly stated that Algorithm uses stochastic
  gradient to gain in scalability. Nevertheless the paper does not
 discuss  reference  [1] dedicated to unbalanced minibatch optimal
  transport (UOT) that can be a good alternative to the proposed method. [1]
  established theoretical properties of UOT in presence of outliers,
  in terms of gradient regularity hence the convergence property. How
  the proposed method connects to [1]?

- The transformation $\mathcal{T}$ defined as an affine-like
  deformation model is not fully explained nor justified.

***Other comments***
- How the final point set registration is obtained ? Does it require computing
  the optimal transport map ? If so, how is it derived from the
  prima-dual equation?
- Report the means and the variances of the performances (MSE, computation time)
  over the runs. That helps the reading of the results.
- In Eq. (9), write $\mathcal{T}_\theta(y_j)$ rather than $y_j$.
- In EQ. (19) $\mathbf{G}$ should rather be $\mathbf{G}_\rho$.
- In section 5.2 the number of samples is sufficiently reasonable to
allow to compare primal solutions to dual ones in term of performances
and computation time. This might highlight the efficiency brought by
the use of the dual form and better illustrates the trade-off
efficiency vs precision.

***Reference***

[1] Fatras, Kilian, et al. "Unbalanced minibatch optimal transport; applications to domain adaptation." International Conference on Machine Learning. PMLR, 2021.

**After rebuttal**

I read authors rebuttal. They address some of the  raised points by  the review (use or not of stochastic gradient to gain in scalability, confidence interval of reported performances, better justification of the transformation $\mathcal{T}$, gradient computation when relaxing the Lipschitz constraint...).

The strong point of the paper is the  KR dual form derivation of $\mathcal{L}_{M,m}$ and this is not sufficiently highligthed in the paper. The authors should consider emphasizing more on that and better  highlight the novelty of the theoretical  KR dual form. During the discussion, the authors clarify how their approach connects to mini-batch regularized Unbalanced OT as promoted in Fatras et al. (2021) that solves the UOT problem in the primal.  This is also a good point.

Overall the paper is  rather dense with almost 46 pages, the appendices  included. Nevertheless, provided aforementioned facts, I update my rating of the paper from 5 to 6.



**Summary Of The Paper:**

The paper deals with  non-rigid point set registration via partial
distribution matching. The problem is formulated as
computing partial Wasserstein-1 (PW) discrepancy between distributions of
the reference point set and of the transformed source point set.  As a
main feature, the paper derives the Kantorovich–Rubinstein dual form
of the PW problem hence, allowing to lift the method to a large scale
setting. Overall the proposed method results in a min-max
problem (similar to Wassertein GAN) to learn the transformation function which minimizes the PW
discrepancy. Empirical evaluations complete the paper.

**Summary Of The Review:**

The paper proposes a theoretical and methodological framework to
achieve robust  point set registration using scalable partial
Wasserstein based on the dual formulation. Their derivations somehow
derive from existing works. To be efficient, the  algorithm
relaxes the
Lipschitz constraint hence, the overall algorithmic scheme lacks
theoretical  guarantees on the matched distribution. Also the paper
lacks convergence, time complexity analysis of computing PW. Finally
connections to works related to minibatch unbalanced optimal
transport that can be competitors of the method are missing.

---

> ### Author Response · Authors · 2021-11-19
> **Response to reviewer x4db (part 1)**
>
> We thank the reviewer for the valuable comments. We address the concerns below.
> 1. **...In that regard, the KR dual derivations of both PW problems appear issued directly from the aforementioned works.
> This renders the main theoretical results of the paper rather incremental.''**
>
> Let us clarify our theoretical contributions.
> Our major theoretical contribution is on $\\mathcal{L}_{M,m}$,
> for which we not only derived the KR form and but also discuss the properties,
> including the existence of the solutions to the KR form,
> the differentiability of the KR form and the ''flatness'' of the potential.
> The other contributions include the investigations of connections between various related formulations of
> $\\mathcal{L}\_{M,m}$,
> $\\mathcal{L}\_{D,h}$ and
> $\\mathcal{W}\_1$.
> These results connect our method and WGAN,
> and they generalize the classic OT theories.
>
> As for the KR formulation of $\\mathcal{L}\_{M,m}$,
> we wish to stress that although $\mathcal{L}_{M,m}$ is closely related to $\\mathcal{L}\_{D,h}$,
> deriving the exact KR form of $\\mathcal{L}\_{M,m}$ is non-trivial,
> due to the difficulties of determining the corresponding $h$.
> This problem is solved by exploiting the dual forms of $\\mathcal{L}\_{M,m}$ and $\\mathcal{L}\_{D,h}$ in Appx.D.2.
> A more traditional but slightly more complicated way to derive the KR form of $\\mathcal{L}\_{M,m}$ is to exploit its dual form using the proposed tool $\\bar{c(h)}$-transform.
> This is briefly discussed in the remark in Appx.D.2.
>
> We have revised the first contribution in the main text to make our contribution clearer:
>
> >Theoretically, we **derive the KR duality for the mass-type PW discrepancy**. We investigate the properties of mass-type PW discrepancy and the closely related distance-type PW discrepancy. In particular, we show their gradients can be efficiently computed, thus they can be optimized via gradient descent.
>
> In addition,
> We have added a new subsection Appx.A.3 providing the detailed theoretical contributions.
>
> 2. **''the gradient computation is biased and results in the loss of theoretical guarantees on the retrieved solution T of the point set registration problem''**
>
> We indeed soften the Lipschitz-1 constraint of the potential by a Lagrangian multiplier acting on the gradient norm,
> thus the loss value may not represent the exact PW discrepancy because the Lipschitz-1 constraint is not necessarily satisfied.
> However,
> such penalty does not introduce bias into our algorithm.
> Because if our algorithm finds an optimal Lipschitz-$k$ function where $k$ may not equal $1$,
> then the estimated PW value is simply $k$ times the true value,
> and so is the gradient,
> i.e.,
> the direction of the gradient is not changed,
> thus no bias is introduced in the optimization problem.
>
> We note that this gradient penalty treatment is not one of our contributions.
> It is proposed in Gulrajani et al., 2017 for WGAN,
> where one can find more justifications of it.
>
>
> 3. **''The paper lacks theoretical analysis ...the comparison of the primal and dual solutions (computation time, precision...) or the empirical convergence illustration...''**
>
> We agree that the theoretical analysis of the speed of our algorithm is important,
> and we leave it to future research.
> In our paper,
> we have provided some experimental results.
> Some statistics of the training process can be found in Appx.F.8,
> where we present the loss, $h$ and the gradient norm as a function of training step.
> The training time of our algorithm is shown in Fig. 8.
>
> In order to compare the primal and KR form,
> we have revised Appx.F.2 to compare the primal and KR form numerically on a simple example.
> We show the our computation in KR form is sufficiently precise (relative error less than 0.2%) for our applications.
> Nevertheless,
> we wish to stress that since our approach relies on the NN which can be adjusted arbitrarily,
> its efficiency and accuracy are not fixed.
> Thus it is hard to rigorously compare our approach to the other primal solvers.

---

> > ### Comment · Reviewer_x4db · 2021-11-29
> > **On the rebuttal 2/2**
> >
> > Regarding the proposed dual forms, here are some comments:
> > * the dual of
> > $\mathcal{L}_{D,h}$
> > derives from previous works (Lellmann et al., 2014; Schmitzer & Wirth, 2019).
> >
> > Hence the one of
> > $\mathcal{L}_{M,m}$
> >
> > might come out using convexity argument. This renders the result on the dual of
> > $\mathcal{L}_{M,m}$
> >
> > hinging heavily on (Lellmann et al., 2014; Schmitzer & Wirth, 2019). Of course the Lagrange parameter related to the mass constraint in
> >
> > $\mathcal{L}_{M,m}$
> >
> > has to be determined through the optimisation procedure.
> > * The gradient  of the dual derives from the well-known envelope theorem provided differentiability of the dual function. One main point of the paper is to establish such differentiability.
> > * Approach such as Fatras et al. (2021) that solves the entropic regularized unbalanced OT using minibatches may have high memory footprint. Nevertheless the approach comes with some guarantees on the computed unbalanced discrepancy. Such type of guarantees may be lost in the proposed method because of the Lipschitz relaxation and the incomplete optimization of the potential network (when $u$ is a small value).

---

> > > ### Author Response · Authors · 2021-11-30
> > > **Further clarification**
> > >
> > > We thanks the reviewer for the time and effort.
> > > We address the concerns as follows.
> > > 1. About the derivation of $\\mathcal{L}\_{M,m}$.
> > >
> > > We do not know what a trivial ''convexity argument'' could be,
> > > but we guess the reviewer is indicating the strong Lagrange duality which allows to switch the sup and the inf in the Lagrangian.
> > > In that case,
> > > we must clarify that Lagrange duality generally does not apply to our problem because the transport plan $\\pi$ is a Radon measure on a continuous space,
> > > thus it is not necessarily a matrix of finite size.
> > > A rigorous treatment is to use Fenchel-Rockafellar duality instead of the Lagrange duality to describe the relation between  $\\mathcal{L}\_{M,m}$ and  $\\mathcal{L}\_{D,h}$,
> > > and that is exactly what we do in Proposition 3,
> > > and we do not think the computations in the proof is trivial.
> > >
> > > If the Lagrange duality is not what the reviewer means,
> > > please let us know what a simple ''convexity argument'' is.
> > >
> > > 2. About the derivation of the gradient.
> > >
> > > Our proof in Proposition 26 is indeed based on the envelope theorem,
> > > but it also relies on the argument that $\\mathcal{L}\_{M,m}(\\alpha, \\beta\_\\theta)$ is continuous w.r.t $\theta$.
> > > This is not difficult,
> > > but we think it is not trivial.
> > >
> > > 3. About the theoretical guarantees of our method and that of Fatras et al. (2021).
> > >
> > > We have already stated that our method and Fatras et al. (2021) solve different problems,
> > > i.e., Fatras et al. (2021) computes mini-batched entropy-regularized OT problem,
> > > while our approach computes Wasserstein-1 problem,
> > > thus **comparing the theoretical basis of these works is irrelevant to the judgement of the contributions of either of them**.
> > > Nevertheless,
> > > we wish to clarify that it is unfair to say Fatras et al. (2021) is a ''theoretical guaranteed'' work while our work is not.
> > >
> > > For a clearer discussion,
> > > we first explain the theories developed in Fatras et al. (2021).
> > > In Fatras et al. (2021),
> > > the authors assume the computation of $OT\_{\\phi}^{\\tau, \\epsilon}$ and $S\_{\\phi}^{\\tau, \\epsilon}$ is accurate for each batch,
> > > and discuss the properties of the mini-batched objective function.
> > > Specifically,
> > > Theorem 1 suggests the computed value (the limited mini-batched results) can be close to the true value (expectation),
> > > when the number of batches (k) and the size of batches (n) are large.
> > > Theorem 2 suggests that the gradient of the true value (expectation) can be computed and the minus function is regular,
> > > thus the gradient descent algorithm can converge to a stationary point in terms of the generalized gradient.
> > >
> > > Now, let us address the concerns about the comparison between our method and Fatras et al. (2021):
> > >
> > > a) Concern about the theoretical guarantees of our work.
> > >
> > > Since we have shown that our objective function is continuous and differentiable almost everywhere,
> > > the gradient descent algorithm naturally converges.
> > > We think that this is a well-known and commonly used fact,
> > > thus we do not delve into the generalized gradient.
> > > Nevertheless,
> > > one can find a rigorous proof of this property based on Clarke gradient in [1].
> > > In summary,
> > > **both our method and Fatras et al. (2021) compute the gradients and guarantee the convergence of the gradient descent algorithm in theory.**
> > >
> > > As for the guarantee of the influence of the mini-batch,
> > > we do not need it because we do not use mini-batch sampling.
> > >
> > > [1] Davis, D., Drusvyatskiy, D., Kakade, S., and Lee, J. D. Stochastic subgradient method converges on tame functions. Foundations of computational mathematics, 20(1):119–154, 2020.
> > >
> > >
> > > b) Concern about the situations when $u$ is small (the practical computation).
> > >
> > > **Both of our method and Fatras et al. (2021) only focus on the theory for the theoretical computation (the precise computation),
> > > and they do not develope any theory for practical computations (inaccurate computation due to limited resource).**
> > > Note in Theorem 2 in Fatras et al. (2021),
> > > the guarantee of the gradient property is only given for the true value (the expectation).
> > > However,
> > > in practical computation,
> > > when the number of Sinkhorn iterationis small,
> > > (the computation of $OT\_{\\phi}^{\\tau, \\epsilon}$ and $S\_{\\phi}^{\\tau, \\epsilon}$ for each batch is not accurate),
> > > or when the number of batches (k) and the size of batches (n) are small,
> > > **the theoretical error bound of the computations and the guarentee of the convergence are not discussed**.
> > > In our method,
> > > $u$ controls the trade off between accuracy and efficiency,
> > > and we experimentally show this in Fig.8.
> > > For the Lipschitz constraint,
> > > as we stated in previous response,
> > > the relaxation of Lipschitz constraint only scales the discrepancy by a constant,
> > > and it does not influence other aspects of the algorithm such as the convergence.
> > >
> > >
> > > Finally,
> > > please let us know if our rely resolves your concerns.
> > > We are looking forward to your response.

---

> > > > ### Comment · Reviewer_x4db · 2021-11-30
> > > > **On the clarification**
> > > >
> > > > Thanks for the details about the dual derivation and the link to mini-batch UOT. Tthe strong point of the paper is the dual KR derivation of $\mathcal{L}_{M,m}$ and this is not sufficiently highligthed in the paper. The authors should consider to emphasize more on that.
> > > > Thanks also for the discussion with Fatras et al. (2021) that clarifies the link to proposed method. Given that, I will update the review accordingly.

---

> ### Author Response · Authors · 2021-11-19
> **Response to reviewer x4db (part 2)**
>
> 4. **''... Algorithm uses stochastic gradient to gain in scalability.... does not discuss reference [1] dedicated to unbalanced minibatch optimal transport (UOT) that can be a good alternative to the proposed method....How the proposed method connects to [1]?**
>
> No, we do not use mini-batch sampling in our algorithm.
> We use only one batch consisting of all data points.
> To move the source set to the reference set smoothly as a whole,
> it is necessary for the PW discrepancy and the coherence regularizer to act on the whole point sets,
> because otherwise there might be undesired local artifacts.
> In this regard,
> Fatras et al. (2021) can not be directly applied to the point set registration problem.
>
> Our approach is not closely related to  Fatras et al. (2021),
> since they are in two different streams of works:
> Fatras et al. (2021)  belongs to the category of entropy regularized OT solver,
> while our approach is a solver for Wasserstein-1 type problem.
> We have added a new section Appx.B to introduce related OT solvers and to clarify the connections between our approach and the existing ones.
>
> 5. **''How the final point set registration is obtained ?...''**
>
> When the transformation $\\mathcal{T}\_{\\theta}$ is obtained,
> we get the aligned point sets by $\\mathcal{T}\_{\\theta}(Y)$.
> No,
> we do not need the transport map.
>
> 6. **''Report the means and the variances of the performances (MSE, computation time) over the runs...''**
>
> We have added a new paragraph in Appx.F.7 where we display the std of the MSE.
> We do not report the std of computation time because the time of each run is almost the same (the differences are no more than 1 second), which is less than %1 of the total computation time.
>
> 7. **''The transformation T defined as an affine-like deformation model is not fully explained nor justified.''**
>
> Thanks for pointing this out.
> We have added more explanations to the first paragraph in Sec 4.3.
> >According to this definition, $\\mathcal{T}\_{\\theta}$ is the composition of two simple
> transformations: a linear transformation and the adding of an offset vector to each point. Despite its
> simplicity, $\\mathcal{T}\_{\\theta}$ includes several useful transformations as its special case. For example, when $V = 0$
> and $A \\in SO(3)$, $\mathcal{T}\_{\\theta}$ becomes the rigid transformation, and when $A = I$ and $t = 0$, $\\mathcal{T}\_{\\theta}$ becomes the
> “drift” transformation in Myronenko \& Song (2010)
>
> 8. **In Eq. (9), write $\\mathcal{T}\_\\theta(yj)$ rather than yj, In Eq. (19) G should rather be $G_\\rho$**
>
> We believe the current formulation (9) is correct and we have changed G to $G_\\rho$ in Eq.(13)
>
> 9. **''... compare primal solutions to dual ones in term of performances and computation time.''**
>
> Please see our answer 3. We have revised Appx.F.2 to provide a comparison between the primal solver and our approach.

---

> > ### Comment · Reviewer_x4db · 2021-11-27
> > **On the rebuttal 1/2**
> >
> > Thanks to  the authors for providing detailed insights in the proposed approach. Hereafter are some remarks on the response.
> > * Std of the performances: it should be better to include the confidence interval  on the plots rather than on Tables  in Appx  F.7. This  might help  the read of the results.
> > * Similarly, comment on  Fatras et al. (2021) may be moved to Section 2  when it comes to scalable partial or unbalanced OT.

---

> > > ### Author Response · Authors · 2021-11-28
> > > **Thanks for the reply**
> > >
> > > We thank the reviewers for the suggestions.
> > >
> > > Since we can not update the draft in the current rebuttal stage,
> > > we report the revisions as follows.
> > > We will incorporate these revisions into the final version of our paper.
> > >
> > > 1. We change Fig.7 in the paper to error plot,
> > > showing the medians in the original Fig.7 and the std reported in Tab.4 and Tab.5 in the appendix.
> > > The format of the new figure is similar to Fig.2 in Ma et al., 2013.
> > >
> > > Jiayi Ma, Ji Zhao, Jinwen Tian, Zhuowen Tu, and Alan L. Yuille. Robust estimation of nonrigid
> > > transformation for point set registration. In 2013 IEEE Conference on Computer Vision and Pattern
> > > Recognition, pp. 2147–2154, 2013
> > >
> > > 2. We cite Fatras et al., 2021 and add a reference to the detailed discussion in the appendix in Sec.2 .
> > > We modify the last paragraph of Sec.2 as follows:
> > > >......they are generally not scalable to large distributions as they require to compute the correspondence between distributions,
> > > or rely on the mini-batch sampling techniques (Fatras et al., 2021),
> > > thus they are not suitable for the registration problem considered in this paper.
> > > The distance-type PW discrepancy.... continuous space.
> > > More discussions of the related computational approaches of Wasserstein type discrepancies can be found in Appx.B.

---

> > > ### Author Response · Authors · 2021-11-28
> > > **Is there any comment on part 2 of the rebuttal?**
> > >
> > > Since the title of the reponse is ''On the rebuttal 1/2'',
> > > we wish to know whether the reviewer has any concern on the 2/2 part of the rebuttal.
> > >
> > > We are looking forward to your constructive comments.

---

> ### Author Response · Authors · 2021-12-01
> **Thanks for your suggestions**
>
> We thanks the reviewers for the helpful suggestions.
>
> We agree that we should highlight the KR form of $\\mathcal{L}\_{M,m}$.
> To this end,
> we revise the appendix to focus on $\\mathcal{L}\_{M,m}$.
> We also re-organize the chapter in appendix,
> where the materials discussing the differentiability (including the existence of maximizers,
> the computation of gradient, and the application to point set) are now  in Appx.D following the formulation section Appx.C,
> and all other properties not directly related to the main theorems (nevertheless are important) are move to the next section Appx.E.
>
> We think such modification makes the contribution clearer and more concise,
> and makes the paper more readable.
>
> Thanks again for your time and helpful discussions.

---

### Author Response · Authors · 2021-11-19
**Change list**

We thank all reviewers for their valuable feedbacks. We have revised our paper according to the feedbacks in the following aspects:

- To make our theoretical contributions clearer, we revise our first contribution in the main text, and we add a new section Appx.A.3 providing the detailed theoretical contributions.
- We add a new experiment on 3D human dataset in Sec.5.5, where we show that our method can handle complicated deformations and incomplete data.
- We add a new section Appx.B to discuss some related computational OT methods.
- We revise section Appx. F.2 to provide a simple example comparing the primal form and the KR form. We provide visualizations of the learned KR form and we show that the KR formulation is sufficiently precise for our application.
- We revise the last paragraph in Sec.4.4 to present simple example to provide an intuition for our algorithm, and we move the discussion about WGAN to the appendix.
- We revise the last paragraph in Sec.3 to clarify that the two discrepancies used in our paper are not equivalent objective functions.
- We revise the last paragraph in Sec.5.2 to make the toy example clearer.
- We add a paragraph discussing the choice of parameters in Appx.F.4.
-  We report the standard deviation of the results in Sec.5.3 in Appx.F.7, where we show that our method is generally more stable than the baseline methods when the outlier ratio is high.

---

### Author Response · Authors · 2021-11-29
**Please can you let us know if you've read our rebuttal and whether we addressed your concerns?**

Dear Reviewers and ACs,

Thank you for your efforts in reviewing or processing our paper.

We have submitted the response letters to address your concerns and we are waiting for your feedback for 10 days. However, up to now, there are only a few post discussions.

Please can you let us know whether you've read our rebuttal and whether we addressed your concerns?

If we did not, please let us know what we failed to address appropriately.

Thanks again for your hard work!!

Authors of Paper

---

### Decision · Program_Chairs · 2022-01-20

**Decision:**

Accept (Poster)

**Comment:**

In this paper the authors proposes to use partial optimal transport to align point cloud in the presence of noise and partially observed data. To this end they express the partial Wasserstein Kantorovich-Rubinstein duality and use it to adapt the classical WGAN loss to partial OT. The optimal alignment between point clouds is then done by minimizing their proposed loss where the dual potential are modeled as deep neural network hence approximating the partial Wasserstein while being solved using mini-batches. Experiments show the interest of the method on a few well understood examples with ground truth from the Stanford repository.

The paper had originally borderline scores with some reviewers concerned about the theoretical contribution. The authors did a very good reply that that greatly appreciated by the reviewers, the one with the lowest score deciding to increase it. The new numerical experiments on the 3D human dataset were also appreciated. The consensus during the discussion was that the paper is worth accepting but that the authors should take into account the comments form the reviewers and better explain their contribution on the KR duality.

For these reason the AC recommends to accept the paper but urges the authors to take into account the comments from the reviewers. In particular the authors should better highlight their theoretical contribution and explain the link and differences with unbalanced minibtach OT.